# Rethinking Dataset Quantization: Efficient Coreset Selection via Semantically-Aware Data Augmentation

**Yangze Liu** *yangze2@illinois.edu*
*University of Illinois Urbana-Champaign*

**Hong Liu**[*] *hlynn@xmu.edu.cn*
*Xiamen University*

**Reviewed on OpenReview:** *https://openreview.net/forum?id=Mb2nn1yx66*

**Code:** *github.com/YangzeLiu/DQ_v2*

## Abstract

Coreset selection aims to reduce the computational burden of training large-scale deep learning models by identifying representative subsets from massive datasets. However, existing state-of-the-art methods face a fundamental accessibility dilemma: they either require extensive training on the target dataset to compute selection metrics, or depend heavily on large pre-trained models, undermining the core purpose of coreset selection in resource-constrained scenarios. Dataset Quantization (DQ) avoids full dataset training but relies on expensive pre-trained models, introducing computational overhead and domain-specific biases that limit generalization. In this work, we comprehensively redesign the DQ framework to establish a more accessible and domain-robust paradigm for coreset selection. Through rigorous analysis, we identify that: (1) MAE functions primarily as biased data augmentation leveraging memorized ImageNet patterns; (2) MAE benefits ImageNet-related datasets but harms out-of-distribution performance; (3) the original pipeline suffers from feature inconsistency between selection and training phases. We propose DQ_v2, which: (1) eliminates pre-trained model dependencies via Semantically-Aware Data Augmentation (SDA) using randomly initialized CNNs; (2) restructures the pipeline by performing augmentation before selection, ensuring feature consistency. Extensive experiments demonstrate that DQ_v2 achieves superior performance across diverse domains (such as ImageNet-1k, CUB-200, Food-101, and medical imaging) while reducing end-to-end coreset construction cost by 41% on ImageNet-1k (95% in the augmentation phase alone), establishing a robust and practical solution for resource-constrained scenarios.

## 1 Introduction

Deep learning has become the gold standard for many computer vision and machine learning tasks (Dosovitskiy et al., 2021), which have seen rapid growth due to increasing model sizes and dataset volumes. However, training emerging deep models, e.g., vision transformers (ViTs) (Dosovitskiy et al., 2021), on large-scale datasets like ImageNet (Deng et al., 2009) and LAION (Schuhmann et al., 2021) requires substantial computational resources, including high-performance GPUs, large memory capacity, and high-speed storage (Bartoldson et al., 2023). These requirements pose a significant barrier to entry for many researchers and practitioners, especially those in resource-constrained environments. Thus, efficiently training large-scale deep learning models with limited resources has become a common concern in both academia and industry.

---

[*]Corresponding author.
This work made limited use of a large language model (LLM) to assist with writing.

Recent research has shown that large-scale datasets contain many redundant and irrelevant samples (Xia et al., 2024; He et al., 2024), which can be compressed into smaller representative subsets without losing model performance. Coreset selection and dataset distillation, as crucial methods to address this issue, aim to choose or synthesize representative subsets from large-scale datasets to reduce computational complexity while maintaining model performance (Guo et al., 2022; Bartoldson et al., 2023). However, **existing coreset selection methods face a fundamental dilemma**: they either require full or partial training on the target dataset to compute selection metrics, or they depend heavily on large pre-trained models, undermining the very purpose of coreset selection—reducing computational burden in resource-constrained scenarios.

**The computational accessibility problem.** Recent state-of-the-art coreset selection methods exhibit severe practical limitations. Methods like $D^2$ Pruning Maharana et al. (2023) and CCS Zheng et al. (2023) require training on the complete target dataset to compute forgetting scores or AUM (Area Under the Margin) scores, while MoSo Tan et al. (2023) demands training a surrogate network for 50 epochs to observe full training dynamics. This creates a paradoxical situation: coreset selection methods designed to reduce training costs actually require substantial upfront computational investments that many practitioners cannot afford.

Dataset Quantization (DQ) (Zhou et al., 2023; Zhao et al., 2024) represents progress by avoiding full dataset training but introduces a different dependency on large pre-trained models, particularly a Masked Autoencoder (MAE) (He et al., 2022). This dependence introduces two critical problems: (1) *Computational overhead* from the MAE model's substantial parameters; (2) *Domain-specific biases* where ImageNet pre-training benefits related tasks but can harm out-of-distribution performance, limiting generalizability.

These limitations motivate a fundamental question: *Can we develop a coreset selection method that is both computationally accessible and free from pre-training dependencies while maintaining or exceeding state-of-the-art performance across diverse domains?*

**In this paper, we answer this question affirmatively by comprehensively redesigning the DQ framework.** Through rigorous empirical and theoretical analysis, we reveal that: (1) MAE functions primarily as biased data augmentation, leveraging memorized ImageNet patterns rather than general image understanding; (2) MAE benefits ImageNet-related datasets but harms out-of-distribution performance; (3) the original DQ pipeline suffers from feature inconsistency between selection and training phases.

**Based on these findings, we propose DQ_v2**, a comprehensively redesigned framework that establishes a new practical paradigm for coreset selection with improved accessibility and domain robustness. Our key contributions are:

- **Systematic problem identification**: We rigorously analyze fundamental limitations in DQ including MAE's distribution-specific overfitting, feature inconsistency in the pipeline, and broader accessibility problems in coreset selection methods.

- **Novel augmentation strategy**: We develop Semantically-Aware Data Augmentation (SDA) using randomly initialized CNNs to preserve semantic objects while diversifying backgrounds, eliminating pre-trained model dependencies.

- **Comprehensive pipeline redesign**: We restructure the framework to perform augmentation before selection, ensuring feature consistency and enabling superior coreset quality without external knowledge dependencies.

- **Establishing practical accessibility**: Extensive experiments on ImageNet-1k, CUB-200, Food-101, and medical imaging demonstrate superior performance with 41% end-to-end cost reduction (95% in the augmentation phase), improved cross-domain generalization, and robustness to distribution shifts.

**Scope.** DQ_v2 is designed for image classification tasks where each image contains a dominant foreground object, an assumption that holds for the majority of standard classification benchmarks (e.g., ImageNet, CUB-200, Food-101). The underlying foreground–background separation relies on the texture-sensitivity inductive bias of CNNs, which may be less effective for tasks where the foreground concept is ill-defined, such as texture classification or satellite imagery. Extending the method to multi-object or scene-level tasks remains an interesting direction for future work.

**Paper organization**: The rest of this paper is organized as follows. Section 2 reviews related work on coreset selection and data augmentation. Section 3 introduces the problem formulation and briefly reviews the original DQ method. Section 4 presents our critical analysis of DQ's limitations from both empirical and theoretical perspectives. Section 5 introduces our proposed DQ_v2 method. Section 6 presents experimental results and analysis. Section 7 concludes the paper with a discussion of limitations and future work.

## 2 Related Work

### 2.1 Coreset Selection and Data Pruning

Coreset selection is a crucial technique for reducing the computational complexity of deep learning models by selecting a representative subset from large-scale datasets.

Early efforts explored various selection criteria, including geometry-based methods (Agarwal et al., 2020; Chen et al., 2012; Sener & Savarese, 2018), uncertainty-based methods (Coleman et al., 2019), error-based methods (Toneva et al., 2019; Paul et al., 2021), decision boundary-based methods (Ducoffe & Precioso, 2018; Margatina et al., 2021), gradient matching-based methods (Mirzasoleiman et al., 2020; Killamsetty et al., 2021), and submodularity-based methods (Iyer et al., 2021).

Recent state-of-the-art methods have achieved impressive performance but at the cost of practical accessibility. $D^2$ Pruning Maharana et al. (2023) utilizes message passing over a dataset graph to jointly consider sample diversity and difficulty, but it requires training on the complete target dataset to compute forgetting scores. Similarly, Coverage-centric Coreset Selection (CCS) Zheng et al. (2023) balances data coverage and importance by computing AUM (Area Under the Margin) scores, which also necessitates full dataset training. Moving-one-sample-out (MoSo) Tan et al. (2023) evaluates each sample's impact on empirical risk but demands training a surrogate network for 50 epochs to observe complete training dynamics. InfoMax Tan et al. (2025) formulates coreset selection as a discrete quadratic programming problem that jointly accounts for individual sample information and pairwise redundancy and solves the resulting quadratic-form objective using an iterative optimization procedure. Mind the Boundary (BoundarySet-CCS variant in our comparisons) Yang et al. (2024) selects samples to reconstruct the decision boundary learned on the full dataset, achieving strong performance but requiring initial full dataset training to establish the reference boundary. Other methods include Moderate coreset Xia et al. (2023), which selects samples with scores close to the median, and AdaPruner Liu et al. (2021), which jointly prunes training data and fine-tunes models.

While these methods achieve strong performance, their requirement for full or extensive partial training on the target dataset creates a fundamental accessibility barrier: researchers with limited computational resources—the very users who would benefit most from coreset selection—often cannot afford the upfront computational investment these methods require.

### 2.2 Dataset Quantization

To address the scalability and accessibility challenges, Dataset Quantization (DQ) (Zhou et al., 2023) was proposed as a method that avoids the need for full dataset training. DQ combines coreset selection with data compression techniques, effectively selecting representative subsets from large-scale datasets while maintaining high performance under various data keep ratios. By using pre-computed features and avoiding iterative training-based selection, DQ represents an important step toward practical coreset selection.

However, DQ's efficiency comes at a different cost: heavy dependence on large pre-trained models. The framework relies on a Masked AutoEncoder (MAE) (He et al., 2022) with a ViT-Large architecture (304M parameters) for image reconstruction, and a pre-trained ResNet model for feature extraction and importance scoring. While these pre-trained models enable DQ to avoid target dataset training, they introduce substantial computational overhead and, more critically, potential domain-specific biases from ImageNet pre-training. Our analysis reveals that MAE's benefits are dataset-dependent: it helps ImageNet-related datasets but can harm performance on out-of-distribution domains. Furthermore, directly removing these pre-trained components leads to performance degradation and increased variance, suggesting they play a crucial but poorly understood role.

In this work, we systematically investigate the role of pre-trained models in DQ and comprehensively redesign the framework to eliminate these dependencies while achieving superior performance. Our approach addresses both the computational accessibility problem and the domain generalization limitation, establishing a new practical paradigm for coreset selection that requires no pre-trained models, avoids training-dynamics-based scoring (e.g., forgetting/AUM) on the target dataset, and demonstrates robust performance across diverse domains.

## 2.3 Data Augmentation

Data augmentation (Shorten & Khoshgoftaar, 2019) plays an essential role in improving model robustness and generalization ability. Traditional data augmentation methods focus mainly on simple image transformations, such as rotation, flipping, and color adjustment. Recent studies have explored more advanced data augmentation strategies, such as random erasing (Zhong et al., 2020), Mixup (Zhang et al., 2018), CutMix (Yun et al., 2019), and "Copy and paste" (Dwibedi et al., 2017; Ghiasi et al., 2020). These methods have achieved significant success in enhancing the performance and stability of vision models. Although these data augmentation methods have achieved significant success in improving model performance, they generally lack consideration of image semantic structure. Cao & Wu (2022) propose a self-supervised learning framework that leverages the inductive bias of random CNNs to preserve semantic objects while mixing up the background, which we repurpose for data augmentation. How to design data augmentation strategies that can both maintain image naturalness and effectively enhance model learning ability remains an open question. In this work, we first observe that the pre-trained MAE model is actually equivalent to a data augmentation method, which introduces prior knowledge and implicit regularization into the training process. Thus, this observation motivates us to explore a new data augmentation strategy that can replace the MAE model in the DQ method.

# 3 Preliminaries

## 3.1 Problem Formulation

Suppose that we have a large dataset $\mathcal{D} = \{(x_i, y_i)\}_{i=1}^{T}$, where $x_i$ is the $i$-th image and $y_i$ is the corresponding label, and $T$ is the total number of training samples. Coreset selection aims to choose an optimal small subset $D_S$ from a large-scale dataset $\mathcal{D}$, where $D_S \subset \mathcal{D}$ and $|D_S| \ll |\mathcal{D}|$. The model trained on $D_S$ can achieve comparable performance to the model trained on the entire dataset $\mathcal{D}$. Finally, the model trained on the coreset $D_S$ can be used to make predictions on the test set.

## 3.2 Review of Original DQ Framework

As discussed above, most coreset selection and dataset distillation methods suffer from some obvious drawbacks, such as poor generalization and low scalability. Therefore, Zhou et al. (2023) proposed DQ, which consists of three main steps: 1) dataset bin generation, 2) selection of subset bin, and 3) image pixel quantization.

The first step aims to generate multiple non-overlapping dataset subsets (referred to as bins), each containing representative and diverse samples. Here, DQ leverages the traditional coreset selection method, i.e., GraphCut method (Iyer et al., 2021) to select the most representative samples. A pre-trained ResNet model is used to extract features for all images, and the GraphCut score is calculated for each unselected sample when added to the current bin. The second step involves random sampling of the generated bins to form the final compressed dataset. This design introduces additional randomness, contributing to improved model robustness and generalization. The final step is to further reduce storage requirements and enhance data quality. This process involves image patching, importance scoring, patch selection, and image reconstruction. Specifically, a pre-trained ResNet model is first used to compute importance scores for different image patches and guide the selection of informative patches; subsequently, the pre-trained Masked Autoencoder (MAE) decoder is used to reconstruct the complete image from the selected patches. Thus, the original DQ framework relies heavily on two large pre-trained models: a pre-trained ResNet for feature extraction and importance scoring, and a 304M-parameter MAE for image reconstruction.

While DQ achieves state-of-the-art performance on various datasets, especially large-scale datasets like ImageNet, it faces several key challenges in computational efficiency and method stability. As we will demonstrate in Section 4, these limitations stem fundamentally from the heavy reliance on large pre-trained models, particularly the 304M-parameter MAE model used in the pixel quantization step, which introduces both computational overhead and domain-specific biases.

## 4 Rethinking DQ: Problems and Theoretical Flaws

In this section, we present a comprehensive analysis of the original DQ method from both empirical and theoretical perspectives. Our investigation reveals fundamental limitations that motivate the design of our improved framework. We first conduct controlled experiments to understand MAE's role in DQ's performance (Section 4.1), then expose a critical theoretical flaw in the original pipeline design (Section 4.2).

### 4.1 Comprehensive Analysis of MAE's Limitations

**MAE's Claimed Role in DQ.** The original DQ paper (Zhou et al., 2023) justifies the use of MAE primarily for storage efficiency: in the third step (pixel quantization), less-informative patches are removed based on importance scores, and the complete image is reconstructed using MAE (Figure 2). The authors claim this process reduces storage requirements while maintaining image quality through reconstruction.

**Logical Contradictions in the Storage Efficiency Claim.** Our investigation reveals that this storage efficiency justification is untenable for several fundamental reasons: (1) In an era of inexpensive storage but scarce GPU compute, using a computationally demanding MAE model merely to save storage space is counterintuitive. The substantial computational overhead far outweighs modest storage savings, contradicting the core motivation of coreset selection to reduce computational burden in resource-constrained scenarios; (2) The reconstruction process requires temporary storage of approximately $1.75\times$ the original dataset size during processing, directly undermining the storage efficiency claims.

**Our Empirical Investigation.** To understand MAE's actual role, we conducted controlled experiments by removing the pixel quantization step from the original DQ method and directly using the selected images from the second step to train the model. We conducted experiments on CIFAR-10, ImageNette, and CUB-200 datasets with different random seeds and report the mean accuracy and variance in Figure 1.

Our results reveal that MAE's impact varies dramatically across datasets: On ImageNette, removing MAE decreases performance from 72.14% to 69.69% and increases variance. On CUB-200, removing MAE significantly *increases* performance across various selection ratios. On CIFAR-10, a small-scale image dataset, removing MAE only slightly increases accuracy. These mixed results suggest that MAE's primary function is not storage efficiency but rather introducing dataset-specific prior knowledge from ImageNet pre-training.

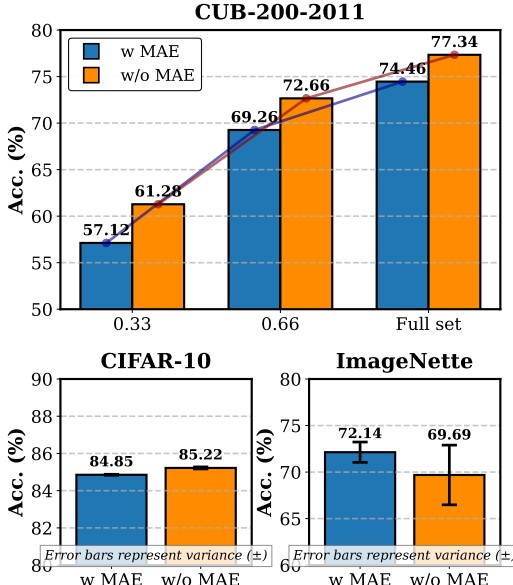

Figure 1: Performance comparison between models with and without MAE pre-training across different datasets. For CUB-200-2011, we evaluate with different data selection rates (0.33, 0.66, and full set).

**Analysis: Four Fundamental Problems with MAE in DQ.** Based on our empirical findings and theoretical analysis, we identify four critical issues:

**Problem 1: Local Interpolation Without Semantic Understanding.** MAE does not truly understand semantic information but performs **local texture interpolation**: filling masked regions with textures from surrounding patches rather than reconstructing semantic content.

Figure 2 demonstrates this behavior under controlled masking experiments on both ImageNet and CUB-200 images. When the foreground object is fully occluded, MAE fills the masked region with textures from the surrounding background rather than recovering the semantic object—a reasonable reconstruction strategy, but one whose effect is highly dataset-dependent.

Specifically, whether this local texture infilling is beneficial depends on the alignment between the image distribution and MAE's ImageNet pre-training. For ImageNet-related datasets, the background smoothing incidentally acts as a mild regularizer, boosting performance (Figure 1, ImageNette). For out-of-distribution datasets such as CUB-200, the same interpolation blurs discriminative fine-grained textures, leading to a performance drop. This asymmetric impact is the core problem: MAE functions as a domain-specific regularizer rather than a neutral compression tool.

**Problem 2: Small-Image Limitation.** MAE performs poorly on small-scale images like CIFAR-10 due to: (1) Ambiguous foreground-background boundaries: In 32×32 images, objects occupy most of the frame, making the CNN's texture-based attention mechanism less discriminative—the entire image becomes uniformly "foreground-like"; (2) Coarse patch granularity: With limited patches available (e.g., 16 patches for 8×8 patch size), dropping any patch risks removing critical information that cannot be reliably reconstructed from sparse neighbors. At this resolution, the spatial information available to any patch-based method is extremely limited, making CIFAR-10 a poor fit for both MAE reconstruction and patch-based foreground separation in general.

**Problem 3: Computational Cost.** The MAE model used in DQ is computationally expensive: (1) The ViT-Large architecture contains 304M parameters; (2) Processing large datasets like ImageNet-1k requires significant GPU resources; (3) This computational overhead contradicts the presumed efficiency goal of Dataset Quantization.

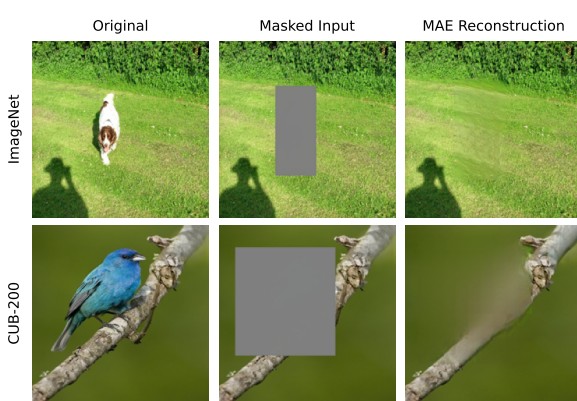

**Problem 4: Fairness in Method Comparison.** When DQ is compared to other coreset selection methods, MAE introduces a confounding factor: (1) Performance improvements may stem from either better coreset selection or the incorporation of ImageNet prior knowledge during reconstruction; (2) This makes it difficult to isolate the true contribution of the coreset selection component; (3) The implicit transfer of knowledge from ImageNet pre-training complicates fair comparison with methods that don't leverage such external knowledge.

Figure 2: Controlled foreground masking on ImageNet (top) and CUB-200 (bottom) shows that MAE fills the missing region with local background texture rather than reconstructing the semantic object.

In summary, our analysis reveals that MAE provides regularization benefits for ImageNet-like datasets by leveraging memorized patterns, but it offers limited value for small-scale or out-of-distribution datasets while introducing significant computational overhead and methodological fairness concerns.

### 4.2 Theoretical Flaw: Feature Inconsistency

Beyond the empirical issues with MAE, we identify a fundamental theoretical problem in the original DQ pipeline itself. In the original DQ method, there exists a critical limitation: the inconsistency between the features used for coreset selection and the features of the images used in the final training. This issue has also been recently analyzed in depth by Zhao et al. (2024). As illustrated in Section 3, DQ follows a sequential

process: first performing dataset bin generation and bin selection based on the original images' features, and then applying pixel quantization with MAE reconstruction.

Following the formal analysis in Zhao et al. (2024), let us denote the original dataset as $D$, and the final output dataset after pixel quantization as $D_{\mathrm{MAE}}$. The GraphCut algorithm used for bin generation calculates submodular gains $G(\mathbf{x}_k)$ based on features extracted from the original dataset $D$:

$$G(\mathbf{x}_k) = \sum_{p \in S_n^{k-1}} \left\| f(p) - f(\mathbf{x}_k) \right\|_2^2 - \sum_{p \in D \setminus (S_1 \cup \cdots \cup S_n^{k-1})} \left\| f(p) - f(\mathbf{x}_k) \right\|_2^2, \tag{1}$$

where $f(\cdot)$ is the feature extractor. However, the model is ultimately trained on images that have undergone MAE reconstruction, where image features have been significantly altered. This means that while the dataset bins are optimized for the original dataset $D$, they may not be optimal for the transformed dataset $D_{\mathrm{MAE}}$ on which the model actually trains.

This inconsistency can lead to suboptimal performance, as the coreset selection process is not aware of the subsequent feature transformations caused by reconstruction. This theoretical flaw, combined with the empirical problems identified in Section 4.1, motivates us to propose a fundamentally redesigned pipeline that addresses both issues simultaneously.

In our improved approach (detailed in Section 5), we perform data augmentation first, then conduct bin generation and selection on the augmented dataset. This reorganized pipeline ensures that feature extraction and GraphCut selection operate in the same feature space that will be used for training. The submodular gains are now calculated as:

$$G(\mathbf{x}_k) = \sum_{p \in S_n^{k-1}} \left\| f(p) - f(\mathbf{x}_k) \right\|_2^2 - \sum_{p \in D_{\mathrm{aug}} \setminus (S_1 \cup \cdots \cup S_n^{k-1})} \left\| f(p) - f(\mathbf{x}_k) \right\|_2^2, \tag{2}$$

where $D_{\mathrm{aug}}$ represents our augmented dataset. This approach ensures that the selected coreset is optimal for the actual feature distribution used during training.

Furthermore, by conducting semantically aware augmentation prior to coreset selection while preserving the original images, we significantly enhance the diversity of the training set. This strategic reordering enables the GraphCut algorithm to select samples from an enriched feature space, thereby identifying the most informative and representative instances. Consequently, our approach can achieve comprehensive coverage of the feature distribution with a minimal number of samples, maximizing information density while minimizing redundancy in the selected coreset.

A detailed formal theoretical framework grounded in submodular optimization theory is provided in Appendix A, which mathematically justifies why our pipeline redesign leads to superior performance.

## 5   Our Proposed Method: DQ_v2

This raises a crucial question: *Can we design a more efficient method that achieves or surpasses DQ's performance without relying on large pre-trained models while also addressing the feature inconsistency flaw?* In this section, we present our answer: Dataset Quantization V2 (DQ_v2), a computationally efficient framework that addresses both the MAE dependency problem and the feature inconsistency issue.

### 5.1   Semantically-Aware Data Augmentation (SDA)

As analyzed in Section 4.1, MAE's reconstruction process in the pixel quantization step serves two roles: (1) preserving semantic objects while modifying background regions, and (2) introducing regularization through reconstruction-based augmentation. However, MAE suffers from distribution-specific overfitting and computational overhead.

This motivates us to design a more efficient data augmentation strategy that achieves similar benefits without pre-trained model dependencies. Classical augmentation methods like CutMix (Yun et al., 2019) generate

---

**Algorithm 1:** Foreground Mask Generation via Randomly Initialized CNN

---

**Input:** Image $x \in \mathbb{R}^{224 \times 224 \times 3}$, grid size $G$ (default 40), foreground ratio $\rho$ (default 0.5)
**Output:** Binary mask $M \in \{0, 1\}^{G \times G}$

1. Initialize ResNet-50 $\phi$ with random weights (no pre-training)

2. Extract layer-4 features: $h = \phi_{\text{layer4}}(x) \in \mathbb{R}^{2048 \times 7 \times 7}$

3. Project via $1 \times 1$ convolution: $h' = \text{ReLU}(\text{Conv}_{1 \times 1}(h)) \in \mathbb{R}^{512 \times 7 \times 7}$

4. Bilinearly interpolate to grid: $h'' = \text{Interp}(h', G \times G) \in \mathbb{R}^{512 \times G \times G}$

5. Channel-wise summation: $a = \sum_{c=1}^{512} h''_c \in \mathbb{R}^{G \times G}$

6. Select top-$\lfloor \rho \cdot G^2 \rfloor$ patches by activation value as foreground ($M_{ij} = 1$); rest as background ($M_{ij} = 0$)

**Return** $M$

---

new samples by cutting and pasting patches between images, but may randomly cut foreground objects, failing to preserve semantic integrity.

Thus, we need to design an augmentation method that: (1) maintains semantic object information, (2) introduces beneficial background variations, and (3) requires no pre-trained models. The foreground–background separation mechanism we employ is based on the work of Cao & Wu (2022), who demonstrated that randomly initialized CNNs exhibit a natural inductive bias—locality and translation equivariance combined with ReLU activations—that produces spatially coherent activation maps highlighting textured, salient regions even without any training. Building on this finding, we identify MAE as a domain-specific regularizer with fundamental limitations in the DQ pipeline (Section 4), and apply this CNN-based mechanism as a more accessible and domain-robust replacement, combined with a restructured pipeline that ensures feature consistency (Section 4.2).

Specifically, the combination of CNN architectures with ReLU activation functions naturally focuses on high-texture regions (foreground objects) while suppressing low-texture regions (backgrounds), enabling automatic semantic object localization without any pre-training. The detailed mask generation procedure is described in Algorithm 1. We repurpose this property for data augmentation: using a randomly initialized CNN to identify semantic objects and then replacing background regions with random patches from other images. This method can effectively maintain the naturalness of the image and introduce beneficial variations. Illustrative examples of this process at different patch granularities ($5 \times 5$, $16 \times 16$, or $40 \times 40$) are provided in Appendix D.1 (Figure 7(a)), where we show that these randomly shuffled background patches contain only texture information, not semantic content. For all experiments reported in this paper, we consistently use the $40 \times 40$ granularity setting, which ensures that the semantic integrity of the main object is preserved while only introducing variation in non-semantic texture patterns.

In addition, we mix the augmented data and the original images to further enhance the diversity of the training data. We also report that using appropriate mixing rates can further improve the model's performance. This strategy offers the following advantages:

- **Semantic Preservation**: By preserving the image's main object region, it ensures that augmented images maintain the original semantic information.

- **Diversity Introduction**: The replacement of background regions introduces new visual contexts, increases data diversity, and improves model generalization.

- **Computational Efficiency**: Compared to using large pre-trained models (like MAE), this method has lower computational overhead and requires no additional model dependencies, making it suitable for resource-constrained environments.

**Theoretical Justification for SDA.** The effectiveness of our SDA strategy can be understood from three complementary theoretical perspectives:

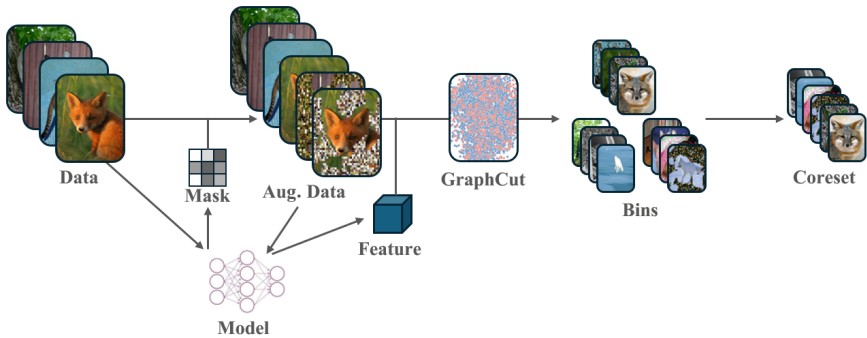

Figure 3: The overall pipeline of DQ_v2.

*(1) Sample Space Expansion Theory:* As formalized in Appendix A (Assumption 2), SDA systematically expands the dataset from $D$ to $D_{\mathrm{aug}}$ with $|D_{\mathrm{aug}}| = 1.5|D|$. This expansion is not arbitrary but semantically structured—each augmented sample $(x'_i, y'_i)$ preserves the label $y'_i = y_i$ while introducing controlled variation in the contextual background. By expanding the sample space before coreset selection, we provide the GraphCut algorithm with a richer pool of candidates from which to select maximally diverse and representative samples (Theorem 1).

*(2) Spurious Correlation Breaking:* Traditional data augmentation often fails to address the problem of spurious correlations between objects and their typical contexts (e.g., soccer balls primarily appearing on grass). Our SDA explicitly breaks these correlations by replacing backgrounds with random patches from other images. This forces the model to learn background-invariant representations that focus on the intrinsic properties of foreground objects rather than contextual cues. By ensuring that semantic objects appear in diverse, unrelated backgrounds during training, SDA prevents the model from relying on spurious background cues for classification.

*(3) Regularization through Controlled Diversity:* Unlike MAE's domain-specific biases (Section 4.1, Problem 1), SDA introduces diversity without injecting external prior knowledge. The random background replacement acts as a powerful regularizer that prevents overfitting to specific background patterns while maintaining semantic integrity. This is particularly valuable when $|D|$ is limited or when the target distribution differs from common pre-training datasets like ImageNet.

**CNN vs. ViT for Mask Generation.** Our deliberate choice of a CNN (ResNet-50) rather than a Vision Transformer for mask generation is grounded in the theoretical analysis of Cao & Wu (2022), who demonstrate that the inductive biases of CNNs—locality and translation equivariance—naturally produce spatially coherent activation maps that highlight textured, salient regions, even without any training. Specifically, in CNNs with ReLU activations, background areas with less texture complexity are progressively deactivated as network depth increases, producing a natural foreground–background separation. Vision Transformers lack these architectural priors (relying instead on global self-attention) and do not exhibit this emergent property.

### 5.2 Our Proposed Framework: DQ_v2

Building upon our SDA strategy and the feature consistency principle discussed in Section 4.2, we propose DQ_v2, a comprehensively redesigned framework that addresses both the pre-trained model dependency and the feature inconsistency issues in the original DQ method.

A key innovation in our approach is the reordering of the pipeline steps: unlike the original DQ, which performs augmentation after coreset selection (leading to feature inconsistency), we perform augmentation first and then selection, ensuring that GraphCut operates in the same feature space used during training (as formalized in Section 4.2, Proposition 1).

Our improved method includes the following key steps: **1) Mask Generation and Data Selection**: We randomly select 50% of the images from the training set for augmentation. For these selected images, we

use a randomly initialized ResNet-50 model to generate foreground masks following Algorithm 1. This step leverages CNN's inductive bias to effectively identify the main objects without requiring any pre-training on object detection tasks (see Section 5.1 for the theoretical justification of why CNNs are preferred over ViTs for this task). **2) Semantically-Aware Data Augmentation (SDA)**: Based on the generated masks, we augment the selected images by retaining their main object parts while replacing the original backgrounds with randomly selected backgrounds from other images. This process maintains the original semantic information while introducing new visual contexts. We then combine these semantically-aware augmented images with the original complete training set, effectively expanding the dataset to 1.5 times its original size with increased diversity. **3) Dataset Binning**: Use an EarlyTrain model—a ResNet-50 trained for only 500 iterations (batch size 256) on the target training set, requiring approximately 3 minutes on Food-101—to extract visual features and then apply the GraphCut method (Iyer et al., 2021) to split the mixed training set, generating multiple non-overlapping bins. The EarlyTrain model provides basic learned representations for computing pairwise similarities needed by GraphCut. Unlike the 304M-parameter pre-trained MAE that requires external ImageNet pre-training, EarlyTrain trains directly on the target data for only 500 iterations, adding negligible overhead. By performing this step after data augmentation, we ensure feature consistency between selection and training. This step ensures that the selected samples are representative and diverse, keeping the core advantages of the DQ method. **4) Bin Sampling**: Randomly select a proportionally adjusted percentage of images from each bin to form the final coreset. Since our dataset has been expanded to 1.5x its original size, we accordingly adjust the selection ratio by a factor of 1/1.5 to maintain the same effective number of samples as other methods. For example, to obtain a coreset equivalent to 60% of the original dataset size, we select 40% (= 60% / 1.5) from the augmented dataset, yielding $0.4 \times 1.5n = 0.6n$ samples, where $n$ is the original dataset size. This adjustment ensures fair comparison with other methods (see Appendix D for detailed analysis). This random sampling process further increases the diversity of the data, allowing users to flexibly adjust the proportions of the data to suit different task requirements. **5) Model Training**: Train the model using the selected coreset.

The complete pipeline of DQ_v2 is illustrated in Figure 3. As the coreset contains both original and augmented images, the model can learn richer and more robust feature representations, enhancing model performance and stability.

Through this comprehensive redesign, DQ_v2 simultaneously achieves three key advantages: (1) eliminates externally pre-trained model dependencies, reducing end-to-end coreset construction cost by 41% on ImageNet-1k (95% in the augmentation phase alone; see Section 6.2 for a full breakdown); (2) ensures feature consistency between selection and training through pipeline reordering; (3) improves performance across diverse domains through semantically-aware augmentation. Experimental results (Section 6) demonstrate that DQ_v2 surpasses the original DQ and other state-of-the-art methods while maintaining practical accessibility for resource-constrained scenarios.

# 6 Experimental Results and Analysis

## 6.1 Experimental Setup

**Datasets:** We conducted experiments on multiple datasets, including ImageNette Howard (2019) (a 10-class subset of ImageNet), CUB-200-2011 Wah et al. (2011), and Food-101 Bossard et al. (2014). These datasets cover a wide range of image classification tasks, enabling us to comprehensively evaluate the performance of our proposed method. In addition, we also conducted experiments on ImageNet-1k to further validate the effectiveness of our proposed method compared to the state-of-the-art methods.

**Implementation Details:** We implement our proposed DQ_v2 method using PyTorch and train the models on NVIDIA V100 GPUs. We use the randomly initialized ResNet-50 model as the backbone for SDA. For the dataset bin selection stage, we utilize the EarlyTrain ResNet-50 model as the feature extractor. The number of dataset bins is set to 10 by default. We also use the timm library (Wightman, 2019) for model training across all datasets. For comparisons with the original DQ, we follow *exactly* the same downstream

evaluation protocol as the original DQ method[1]. This ensures that improvements over DQ are attributable to our coreset selection method rather than differences in training procedures. A detailed stability analysis comparing DQ_v2 with the original DQ method across multiple datasets and random seeds is provided in Appendix B.

### 6.2 Comparison with original DQ

In this part, we primarily evaluate the performance of our DQ_v2 method compared to the original DQ method. Specifically, the results on ImageNette are shown in Figure 4 (a), and the comparison on CUB-200-2011 and Food-101 is shown in Figure 4 (b).

The results show that our method achieves better performance compared to the original DQ method. Notably, in the Food-101 dataset, our method achieves a significant performance improvement of 3.98% compared to the original DQ method, while on ImageNette, we observe a gain of 1.66%. The larger performance gap on Food-101 aligns with our analysis: MAE primarily benefits ImageNet-related datasets (such as ImageNette) but functions merely as Gaussian blur for non-ImageNet datasets like Food-101. This explains why removing MAE and using our SDA approach yields more substantial improvements on Food-101.

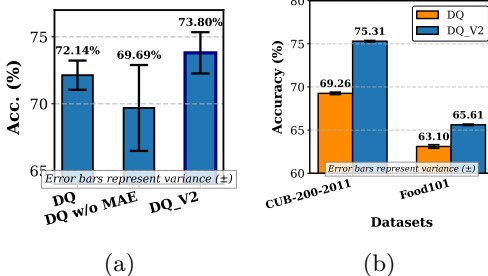

(a)      (b)

Figure 4: Preliminary evaluation of DQ_v2. (a) Results on ImageNette. Means and variances are computed over five runs with different seeds.

Moreover, to demonstrate that our coreset selection generalizes across different architectures, we train a ViT model on the subset selected using the ResNet-50 feature extractor and evaluate the performance on ImageNette. The accuracy of the DQ method is 55.30%±2.73%, and ours is 57.67%±1.20%. The results further verify the effectiveness and generalizability of our proposed method.

**Computational Efficiency.**

Beyond accuracy improvements, our method also offers significant computational advantages. It is worth noting that the original DQ method already provides substantial computational benefits over many alternative approaches. Unlike dataset distillation methods where computational cost scales quadratically with the size of the synthetic set, DQ offers more favorable scaling through its binning approach. Moreover, compared to methods like $D^2$ Pruning Maharana et al. (2023) and AdaPruner Liu et al. (2021) that require training on the complete dataset, DQ's pipeline avoids this computational burden entirely by operating directly on feature representations. Our DQ_v2 method preserves these fundamental efficiency advantages while providing further improvements.

| Pipeline | Stage | Time |
|---|---|---|
| DQ_v2 | Mask generation | 31m 58s |
| | SDA image generation | 2m 59s |
| | EarlyTrain (500 iter) | 3m 14s |
| | GraphCut (on 1.5× data) | 11h 50m 48s |
| | **Total** | **12h 29m** |
| DQ | MAE augmentation | 11h 40m 23s |
| | GraphCut (on 1.0× data) | 9h 36m 32s |
| | **Total** | **21h 17m** |
| End-to-end speedup | | **41.3%** |

Table 1: End-to-end coreset construction cost on ImageNet-1k (1×Xeon CPU + 1×RTX 4090).

Table 1 presents a complete end-to-end cost comparison of the coreset construction pipelines. DQ_v2 reduces total construction time by **41.3%** (12h 29m vs. 21h 17m). The primary savings come from replacing the 304M-parameter MAE (11h 40m) with lightweight SDA generation (35 min total for mask generation and image creation)—a 95% reduction in augmentation cost. Although GraphCut runs somewhat longer on the expanded 1.5× candidate pool (~2h overhead), this is more than offset by eliminating MAE. We note that the EarlyTrain step adds only 3 minutes of overhead. An additional breakdown on Food-101 is provided in Appendix G.

---

[1]For ImageNet-1k experiments, we use ResNet-50 as the backbone model. For other datasets (CUB-200, Food-101, etc.), we use ResNet-18 as the backbone model. This choice follows standard practice for datasets of different scales.

## 6.3 Comparison with State-of-the-art Methods on ImageNet-1k

To fully validate the effectiveness of our proposed DQ_v2 method, we conduct extensive experiments on the large-scale ImageNet-1k dataset, comparing it with recent state-of-the-art coreset selection and data pruning methods, including $D^2$ Pruning, CCS, MoSo, InfoMax, and BoundarySet-CCS. The results are shown in Figure 5. All baseline Top-1 accuracies are taken directly from the corresponding papers' reported results.

The results demonstrate that our DQ_v2 method consistently outperforms other leading data pruning approaches across most data keep ratios (Figure 5). Compared to the original DQ method, DQ_v2 shows consistent improvements across all keep ratios, validating the effectiveness of our comprehensive pipeline redesign and semantically-aware data augmentation strategy.

Against other recent state-of-the-art methods, including $D^2$ Pruning Maharana et al. (2023), CCS Zheng et al. (2023), MoSo Tan et al. (2023), InfoMax Tan et al. (2025), and BoundarySet-CCS Yang et al. (2024), DQ_v2 demonstrates competitive or superior performance, particularly when the data keep ratio exceeds 20%, where it generally maintains a leading position. Notably, at a 60% data keep ratio on ImageNet-1k, our DQ_v2 method achieves an impressive 75.94% Top-1 accuracy. This performance is particularly significant as it surpasses the reported accuracy of CCS (75.58%) which requires a larger data selection rate of 80%. This achievement establishes DQ_v2 as a highly efficient solution for dataset compression on ImageNet-1k, achieving better

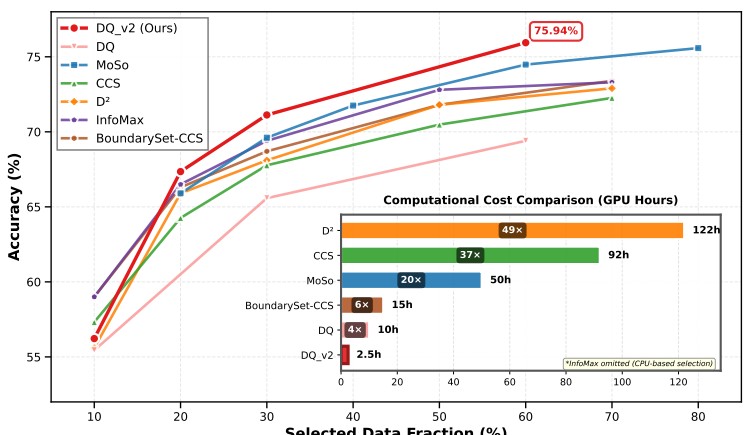

Figure 5: Performance comparison with state-of-the-art coreset selection methods on ImageNet-1k. DQ_v2 achieves competitive or superior accuracy across most keep ratios.[2]

performance with less data while also eliminating the need for expensive pre-trained models and full dataset training that other methods require.

At high keep ratios, all methods naturally approach random sampling because the number of distinct subsets shrinks rapidly, making the gap between selection strategies increasingly small.

At very low data keep ratios (10%), DQ_v2 is outperformed by InfoMax and BoundarySet-CCS. This is expected: DQ_v2 expands the candidate pool from $N$ to $1.5N$ before selection, so at a 10% keep ratio the effective budget over the expanded pool drops to ∼6.7%, and part of the limited budget may be allocated to SDA variants rather than original samples. A more detailed mechanism analysis and a practical cost comparison (showing that at very low ratios, the overhead of sophisticated selection can outweigh its benefit over simpler strategies) are provided in Appendix D.

Further detailed analyses, including the factors contributing to DQ_v2's efficiency and a discussion on the fair comparison of training sample counts, are provided in Appendix D.

## 6.4 Robustness and Cross-Domain Generalization

To comprehensively evaluate DQ_v2's practical applicability and address the question of whether our method truly overcomes the domain-specific limitations identified in Section 4.1, we conduct two additional experiments that test the method's behavior beyond the standard ImageNet evaluation.

---

[2]Results are taken from each method's original publication; some methods do not report results beyond 60–70% keep ratio. Bubble size reflects estimated GPU computation hours for coreset construction, drawn from each method's reported computational requirements under varying hardware; these are heterogeneous estimates and not directly comparable. For a rigorous end-to-end wall-clock comparison of DQ vs. DQ_v2 under identical hardware, see Table 1.

| Ratio | Accuracy (%) | | Gain |
|---|---|---|---|
| (%) | DQ | DQ_v2 | (%) |
| 5 | 81.46 | 88.48 | +7.02 |
| 10 | 85.96 | 90.01 | +4.05 |
| 15 | 86.46 | 92.49 | +6.03 |
| 30 | 87.99 | 93.08 | +5.09 |
| 60 | 90.86 | 95.06 | +4.20 |

Table 2: Cross-domain generalization results on COVID-19 Radiography Database.

**Cross-Domain Generalization.** A central claim of our work is that DQ_v2 avoids the domain-specific biases inherent in MAE-based approaches (Section 4.1, Problem 1). To validate this claim, we evaluate our method on the COVID-19 Radiography Database (Chowdhury et al., 2020), a medical imaging dataset that is significantly different from ImageNet in terms of image characteristics, semantic content, and visual features. This dataset contains chest X-ray images across multiple categories, representing a challenging out-of-distribution scenario.

Table 2 presents the comparison between the original DQ and our DQ_v2 across various data keep ratios. The results demonstrate substantial improvements across all settings, with gains ranging from +4.05% to +7.02%. Notably, even at very low selection rates (5%), DQ_v2 achieves 88.48% accuracy compared to DQ's 81.46%, representing a remarkable 7.02% improvement. This consistent superiority across all keep ratios strongly validates our hypothesis that eliminating pre-trained model dependencies enables better generalization to out-of-distribution domains. The substantial performance gains on this medical imaging dataset directly demonstrate that DQ_v2's design successfully addresses the domain-specific overfitting problem we identified in the original DQ method.

**Robustness to Image Corruptions.** Beyond cross-domain generalization, we evaluate whether models trained on DQ_v2-selected coresets exhibit improved robustness to distribution shifts at test time. We use the ImageNet-C benchmark (Hendrycks & Dietterich, 2019), which applies 15 types of corruptions (e.g., Gaussian noise, motion blur, JPEG compression) at 5 severity levels to the ImageNet validation images, providing a comprehensive assessment of model robustness.

We train ResNet-50 models on a 60% coreset selected by DQ_v2 and evaluate them on all ImageNet-C corruptions, computing the mean Corruption Error (mCE) metric. Our method achieves an mCE of 71.26, compared to 76.7 for the baseline ResNet-50 trained on the full ImageNet dataset (a 5.44 point improvement). This substantial improvement in robustness can be attributed to our SDA strategy: by explicitly breaking spurious correlations between foreground objects and their typical backgrounds (see Section 5.1 for details on SDA), DQ_v2 encourages models to learn background-invariant representations that naturally generalize better to distribution shifts. The coreset selected from the augmentation-enriched space contains more diverse contextual variations, effectively serving as a built-in robustness-enhancing regularizer during training.

These robustness and generalization results provide strong empirical evidence that DQ_v2 not only matches or exceeds DQ's performance on standard benchmarks but also demonstrates superior practical applicability across diverse domains and under distribution shifts. Detailed ablation studies on the impact of key components (bin division algorithms, SDA patch granularities, and mixing ratios) are provided in Appendix C, and comprehensive new ablation experiments (background types, augmentation ratios, component ablation, and comparison with simpler augmentation baselines) are presented in Appendix F.

## 7  Conclusion, Limitations and Future Work

In this paper, we address the fundamental accessibility dilemma in coreset selection: existing state-of-the-art methods either require extensive training on target datasets or depend heavily on large pre-trained models, undermining the core purpose of reducing computational burden in resource-constrained scenarios. Through comprehensive analysis of the Dataset Quantization (DQ) method, we identify critical limitations: (1) MAE

provides dataset-specific benefits through ImageNet prior knowledge but suffers from distribution-specific overfitting, benefiting ImageNet-related datasets while harming out-of-distribution performance; (2) the original pipeline suffers from feature inconsistency between selection and training phases due to applying augmentation after selection; (3) heavy reliance on expensive pre-trained models (304M-parameter MAE and pre-trained ResNet) introduces substantial computational overhead.

To address these fundamental issues, we propose DQ_v2, which establishes a new practical paradigm for coreset selection with improved accessibility and domain robustness. By eliminating all externally pre-trained model dependencies through Semantically-Aware Data Augmentation (Cao & Wu, 2022) and fundamentally restructuring the pipeline to ensure feature consistency, DQ_v2 provides a coreset selection method that: (1) requires no externally pre-trained models, (2) avoids training-dynamics-based scoring on the target dataset, and (3) demonstrates robust performance across diverse domains. Through extensive experiments on diverse domains (ImageNet-1k, CUB-200, Food-101, medical imaging), we show that DQ_v2 improves performance and stability over the original DQ and other state-of-the-art coreset selection methods, while reducing end-to-end coreset construction cost by 41% and improving cross-domain generalization and robustness to distribution shifts.

**Limitations and Future Work.** Despite its strong performance, DQ_v2 presents opportunities for further development: **1.** The current SDA employs a fixed 50/50 proportion for foreground and background patches, which may not be optimal for all images. Future work will focus on developing adaptive techniques to determine this ratio on a per-image basis, potentially improving object localization precision. **2.** Our validation of DQ_v2 has thus far been confined to classification tasks. A key direction for future research is to extend its application and evaluate its effectiveness for other visual recognition tasks, such as object detection and segmentation. **3.** The SDA mechanism relies on the assumption that images contain a dominant foreground object that can be separated from the background via texture-based activation maps. This assumption holds for standard classification benchmarks but may break down for tasks where the foreground concept is ill-defined, such as texture classification or satellite imagery.

**Reproducibility Statement.** To facilitate reproducibility, our code is publicly available at `https://github.com/YangzeLiu/DQ_v2`.

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

# A    Formal Theoretical Framework

To provide a rigorous foundation for our approach, we present a formal theoretical framework grounded in submodular optimization theory. This framework mathematically justifies why our pipeline redesign leads to superior performance.

**Assumption 1: Properties of the Set Function.** Let $\Omega$ be the ground set of dataset elements, $|\Omega| = |D|$, and let $f_\Omega : 2^\Omega \to \mathbb{R}$ be a normalized, monotone, and submodular function as defined in Iyer et al. (2021). We use subscripts (e.g., $f_D$, $f_{D_{\text{aug}}}$) to denote the same function instantiated on different ground sets. In our method, $f$ is instantiated as the generalized graph cut function with similarity kernel $s$ and parameter $\lambda \geq 2$, ensuring monotonicity and submodularity (Lemma 17 in Iyer et al. (2021)).

**Assumption 2: Sample Space Expansion.** Let $D = \{(x_i, y_i)\}_{i=1}^n$ be the original dataset. Our Semantically-Aware Data Augmentation (SDA) produces semantically consistent variants with altered background textures but preserved labels. In our implementation, we augment exactly 50% of the dataset, yielding:

$$|D_{\text{aug}}| = |D| + 0.5|D| = 1.5|D| \quad \Rightarrow \quad \alpha = 1.5. \tag{3}$$

*Label Preservation*: For any $(x_i, y_i) \in D$, and any SDA variant $(x_i', y_i') \in D_{\text{aug}}$ derived from $x_i$, we have $y_i' = y_i$.

**Theorem 1 (Submodular Optimization in Expanded Space).** We recall the submodular mutual information from Iyer et al. (2021):

$$I_f(A; B) = f(A) + f(B) - f(A \cup B). \tag{4}$$

The generalized graph cut function (Lemma 17 in Iyer et al. (2021)) is:

$$f(A) = \lambda \sum_{i \in \Omega} \sum_{a \in A} s_{ia} - \sum_{a_1, a_2 \in A} s_{a_1 a_2}, \quad \lambda \geq 2. \tag{5}$$

GraphCut selects a subset $S \subseteq D_{\text{aug}}$ by maximizing coverage while minimizing redundancy, capturing:

- Original diversity: from $D$

- Synthetic diversity: from SDA

- Boundary coverage: better representation of rare or borderline cases

Since $D_{\text{aug}} \supset D$ and $f$ is monotone, we have:

$$\max_{|S| \leq k} f_{D_{\text{aug}}}(S) \geq \max_{|S| \leq k} f_D(S). \tag{6}$$

This formalizes the advantage of selection in the expanded space.

**Proposition 1 (Feature Space Consistency Advantage).** Let $\phi : \mathcal{X} \to \mathbb{R}^d$ be the feature extractor, and define $F(X) = \{\phi(x) \mid x \in X\}$. Let $F_{\text{sel}}$ and $F_{\text{train}}$ be the feature spaces seen at selection and training time, respectively.

*Augmentation-before-selection (ours):*

1. SDA: $D \to D_{\text{aug}}$

2. Selection: $S \subset D_{\text{aug}}$ via GraphCut

3. Training: model trains on $S$

Then: $F_{\text{sel}} = F(D_{\text{aug}})$, $F_{\text{train}} = F(S) \subseteq F(D_{\text{aug}})$, which implies $F_{\text{sel}}$ and $F_{\text{train}}$ are consistent.

*Augmentation-after-selection (baseline DQ):*

1. Selection: $S \subset D$ via GraphCut

2. Augmentation: $S \rightarrow S_{\mathrm{transformed}}$

3. Training: model trains on $S_{\mathrm{transformed}}$

Then: $F_{\mathrm{sel}} = F(D)$, $F_{\mathrm{train}} = F(S_{\mathrm{transformed}})$, which implies $F_{\mathrm{sel}} \neq F_{\mathrm{train}}$ when augmentation changes feature distributions.

*Conclusion:* Our pipeline ensures that the selection algorithm operates within the same feature distribution as during training, eliminating the distribution mismatch identified in the original DQ.

## B   Stability Analysis

As discussed in the main paper, the pixel quantization step plays an important role in reducing the variance of the trained model. Therefore, in this section, we investigate the stability of our proposed DQ_v2 method compared to the original DQ method.

On the ImageNette dataset (Figure 4 (a) in the main paper), we observe that removing the MAE model from the original DQ method significantly increases variance and decreases performance (from 72.14% to 69.69%). In contrast, our proposed DQ_v2 method achieves comparable variance while obtaining higher accuracy (73.80%). This result indicates that our proposed method can effectively address the stability issue of the original DQ method while maintaining high performance. Furthermore, we observe similar stability improvements on the Food-101 dataset, where our proposed method achieves a variance of 0.0745, significantly lower than DQ's 0.197.

These results underscore the effectiveness of our method in addressing the instability issue of DQ when the pre-trained model is removed. We attribute DQ_v2's stability primarily to the following factors: 1) By employing semantically-aware background replacement, it provides more diverse training samples, reducing dependence on specific background features while expanding the sample space and mitigating the risk of overfitting. 2) Maintaining a balance of original images and semantically-aware augmented images in the dataset preserves the authenticity of the original data while introducing sufficient diversity. 3) Our modified pipeline, which performs data augmentation before coreset selection, prevents the feature shifts that occur in the original DQ method (as discussed in Section 4.2 of the main paper), where data augmentation after coreset selection can lead to instability.

## C   Ablation Studies

In this section, we conduct ablation studies to analyze the impact of key components in our proposed DQ_v2 method.

### C.1   Impact of Bin Division Algorithms

We analyze the impact of different bin split algorithms on the performance of our proposed method. We compare the performance of our method with three different bin split strategies, including GraphCut (Iyer et al., 2021), Random, and Uniform methods, on the ImageNette dataset at various data keep ratios (1%, 2%, and 5%). To ensure reliability, we repeat each experiment three times with different random seeds and report the mean accuracy and standard deviation. The results are shown in Figure 6.

The results demonstrate that the GraphCut method consistently achieves the best performance across all data keep ratios (Figure 6). At the 1% keep ratio, GraphCut achieves 32.7% accuracy, outperforming Random (31.5%) and Uniform (29.4%). This performance advantage becomes more pronounced at higher keep ratios, with GraphCut reaching 60.1% at 5%, compared to 56.8% for Random and 57.3% for Uniform. These results indicate that GraphCut can effectively select the most representative samples from the dataset, which improves the performance of the trained model.

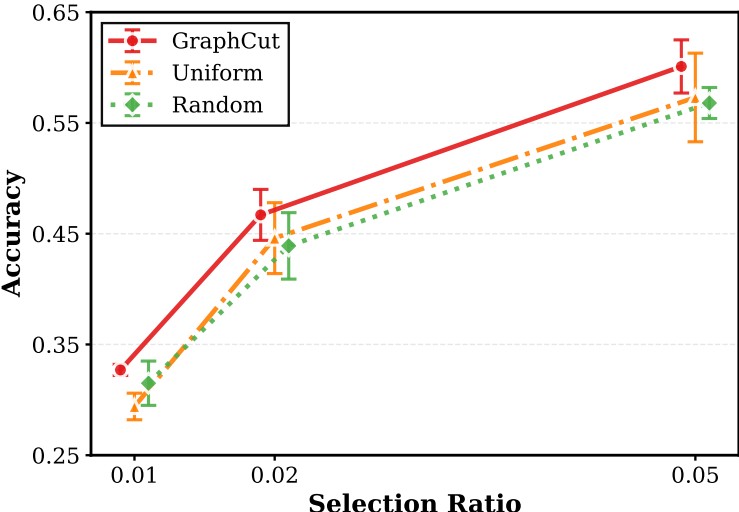

Figure 6: Visualization of bin division algorithm performance on ImageNette.

In summary, the GraphCut algorithm consistently outperforms Random and Uniform methods across different data keep ratios. The performance improvements are substantial, particularly at lower keep ratios where sample selection quality is most critical. Therefore, in practice, we strongly recommend using the GraphCut method to achieve optimal performance and stability.

# D   Detailed Analysis of DQ_v2 Performance on ImageNet-1k

### Performance Analysis at Low Data Ratios

We observe that at extremely low data keep ratios (e.g., 10%), our DQ_v2 method performs marginally below CCS. This is because DQ_v2 expands the candidate pool from $N$ to $1.5N$ before selection; at a 10% keep ratio, the effective budget over the expanded pool drops to ∼6.7%, and part of the limited budget may be allocated to SDA variants rather than original samples, making pool expansion less advantageous when the selection budget is very small. A practical cost analysis comparing sophisticated selection at low ratios against simpler strategies with higher fractions is provided in Appendix H.

### Achieving Superior Efficiency and Performance against SOTA

DQ_v2 can achieve higher accuracy with less data (e.g., 75.94% at 60% keep ratio vs. 75.58% at 80% for the strongest baseline) due to the synergy between SDA and GraphCut. SDA expands the candidate pool with semantically consistent variations, and GraphCut then selects a diverse and informative subset from this enriched space, yielding a better accuracy–efficiency trade-off at moderate keep ratios.

### Fair Comparison of Training Sample Count

One might question whether DQ_v2 actually uses more training samples than other methods at the same reported keep ratio, given that our pipeline includes a data expansion step. To address this concern, we provide a mathematical formulation to demonstrate that our method maintains the same number of training samples as other methods at equivalent keep ratios.

Let $x$ denote the number of training samples in the original dataset. A conventional coreset selection method with a keep ratio of $r$ would select $r \cdot x$ samples. In our DQ_v2 pipeline, we first expand the dataset to $1.5x$ samples through semantically-aware augmentation, and then apply a proportionally reduced selection ratio of

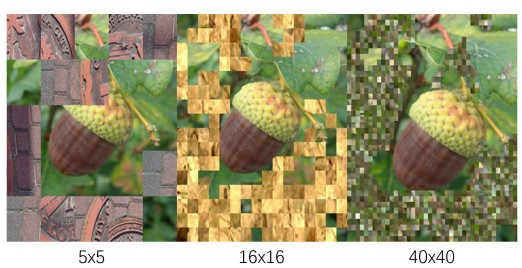

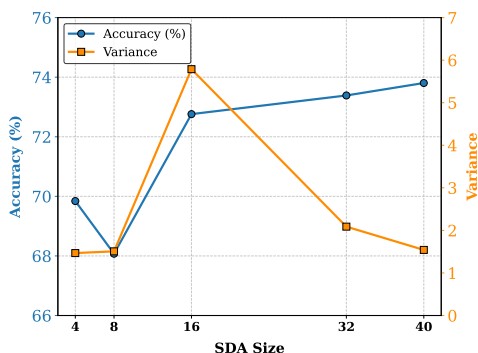

(a) Illustrative examples of different SDA sizes  (b) Accuracy and variance for different SDA sizes

Figure 7: Impact of SDA sizes on DQ_v2 performance on ImageNette.

$\frac{r}{1.5}$ to maintain the same final count:

$$\text{Number of samples} = 1.5x \cdot \frac{r}{1.5} = r \cdot x \tag{7}$$

For instance, to obtain a 60% subset ($0.6x$) from the original dataset of size $x$, we first expand it to $1.5x$ and then select 40% ($0.4 \cdot 1.5x = 0.6x$). Therefore, DQ_v2 uses exactly the same number of training samples as other methods at equivalent keep ratios, ensuring a fair comparison.

## D.1 Impact of SDA Size

We study the SDA patch size, which controls the granularity of patch-wise background replacement. Larger values generally preserve semantic structure better (Figure 7(a)). We evaluate five sizes on ImageNette; results are summarized in Figure 7(b).

SDA patch size strongly affects performance: very small sizes (e.g., 4×4, 8×8) degrade accuracy, while larger sizes improve both mean accuracy and stability, peaking at 40×40.

We attribute this to finer background replacement removing more background-specific cues, which encourages a stronger reliance on the foreground. Based on these results, we use an SDA patch size of 40×40 for optimal performance and stability.

## D.2 Impact of Mixing Ratio

Finally, we investigate the impact of the mixing ratio between the original images and Semantically-Aware augmented images on the performance of our proposed method. Since we do not need to rely on the pre-trained MAE model, we can simply mix semantically-aware augmented data and original images to improve the diversity and quality of the data. Thus, we evaluated the performance of our proposed method with different mixing ratios in the CUB-200, Food-101, and ImageNet-30 datasets. The ImageNet-30 dataset, a subset of ImageNet-1k, was utilized in these specific experiments to enable faster validation.

The results are shown in Figure 8. We observe that performance consistently achieves the best score when we use all original images together with 50% SDA images.

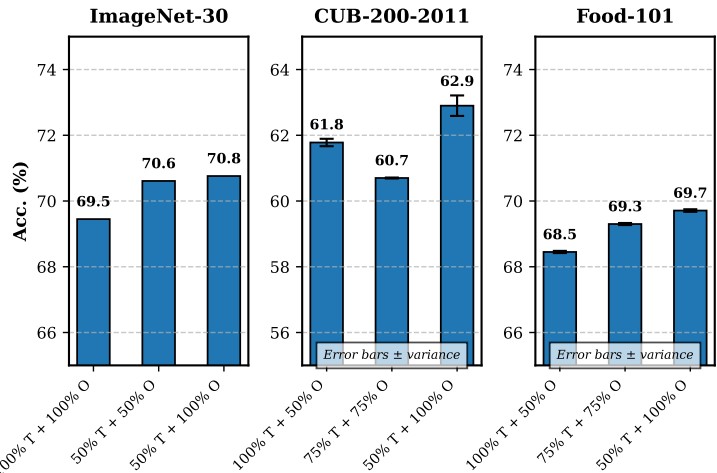

Figure 8: Performance with different T (Semantically-Aware augmented) and O (Original) image ratios. We report the mean accuracy (%) and variance in five runs with different seeds.

# E Experiment Configuration Summary

To address concerns about the consistency of experimental settings across compared methods, we provide a detailed summary of the training configurations used by each method on ImageNet-1k. All numbers are collected from the original publications.

Table 3: **ImageNet experiment configuration summary of compared methods.** Entries are collected from original papers and their appendices. "NR"-style missing details are marked explicitly in text when a paper does not report a single fixed setting.

| Method | Selection / Scoring Setup (ImageNet) | Eval Backbone | Training Hyperparameters for ImageNet Evaluation | Notes |
|---|---|---|---|---|
| DQ_v2 (Ours) | SDA masks from a **randomly initialized ResNet-50**; coreset selection with **EarlyTrain ResNet-50 + GraphCut** on augmented data. 50% SDA augmentation (expanded pool size 1.5×; final keep ratio implemented as $r/1.5$). | ResNet-50 | Follow DQ downstream protocol (timm). In our released ImageNet run logs: SGD (mom.=0.9), cosine schedule, 260 epochs, batch size 128, lr 0.075, wd $2 \times 10^{-5}$, warmup 3 epochs. | No pre-trained MAE dependency. SDA patch size $40 \times 40$ used in final setting. |

*Table 3 continued from previous page.*

| Method | Selection / Scoring Setup (ImageNet) | Eval Backbone | Training Hyperparameters for ImageNet Evaluation | Notes |
|---|---|---|---|---|
| DQ Zhou et al. (2023) | ImageNet bin generation uses ViT-Base features (model pretrained on full ImageNet for 10 epochs); default bin number $N$=10; patch drop ratio $\theta$=25; MAE-based reconstruction in original pipeline. | Architecture-dependent (ResNet / ViT / Swin / ConvNeXt / MobileNetV2) | ImageNet is trained in DDP using **timm default scripts** for each architecture. | Paper does not enumerate a single unified ImageNet optimizer/lr/epoch tuple (depends on architecture recipe). |
| MoSo Tan et al. (2023) | Surrogate network ResNet-50; MoSo score uses training dynamics from a 50-epoch surrogate run with 10 sampled time steps. | ResNet-50 | Paper states settings before/after pruning are kept consistent; algorithm uses SGD formulation. | Detailed ImageNet optimizer/lr/wd/scheduler tuple is not explicitly reported in paper text. Experiments run on 8× Tesla V100. |
| CCS Zheng et al. (2023) | For ImageNet, CCS is applied on AUM. Strata number $k$=50; hard cutoff $\beta$ tuned by grid search (ImageNet best: 30%→0, 50%→0.1, 70%→0.2, 80%→0.2, 90%→0.3). Difficulty scores from full-data training (90 epochs for forgetting/AUM; first 10 epochs for EL2N). | ResNet-34 | 300,000 iterations (about 60 epochs), batch size 256, SGD (mom.=0.9), wd $10^{-4}$, initial lr 0.1, cosine annealing scheduler. | ImageNet comparison uses one run per model due cost. |
| D$^2$ Maharana et al. (2023) | Graph message passing with ImageNet AUM difficulty signal. Best $(k, \gamma_r)$ on ImageNet by pruning rate: 30%:(50,1.0), 50%:(50,1.0), 70%:(100,0.3), 80%:(10,0.1), 90%:(10,0.0). | ResNet-34 | Uses ResNet-34 training hyperparameters from CCS to keep comparisons consistent. | Paper explicitly states following Zheng et al. (CCS) ResNet-18/34 training setup. |
| InfoMax Tan et al. (2025) | Discrete quadratic programming objective combining sample information and pairwise redundancy; weighting coefficients selected on validation set. | ResNet-50 | SGD optimizer; 100 epochs on ImageNet-1K (200 for CIFAR100/Tiny-ImageNet); batch size 256; initial lr 0.2; wd $5 \times 10^{-4}$; cosine annealing scheduler; random crop + horizontal flip. | Implementation reported on 8× NVIDIA A100 GPUs. |

*Continued on next page.*

*Table 3 continued from previous page.*

| Method | Selection / Scoring Setup (ImageNet) | Eval Backbone | Training Hyperparameters for ImageNet Evaluation | Notes |
|---|---|---|---|---|
| BoundarySet CCS Yang et al. (2024) | Boundary-distance based selection plus CCS. For ImageNet distance estimation: step size $10^{-4}$, max step $K$=50. | ResNet-34 | 300,000 iterations, batch size 256, SGD (mom.=0.9), wd $10^{-4}$, initial lr 0.1, cosine annealing scheduler. | CCS grouping over distance bins $d(x,y) \in [0, K]$ to improve coverage at high pruning rates. |

## F  Comprehensive Ablation Studies

We conduct comprehensive ablation experiments on ImageNette at 20% keep ratio with 3 random seeds to analyze the impact of each design choice in DQ_v2.

### F.1  Background Type

We compare three background generation schemes for SDA: **patches** (foreground patches from other images, our default), **Gaussian noise**, and **solid gray**.

Table 4: Impact of background type on DQ_v2 performance (ImageNette, 20% keep ratio, 3 seeds).

| Background Type | Accuracy (%) |
|---|---|
| Gray | $76.47 \pm 2.36$ |
| Patches (default) | $76.67 \pm 1.05$ |
| Noise | $73.93 \pm 2.56$ |

All three variants perform within each other's standard deviations (Table 4), indicating that the method is **robust to the choice of background type**. The foreground preservation—not the specific background content—is the key factor driving SDA's effectiveness. We default to patches as it aligns with the original SDA design motivation and produces the lowest variance.

### F.2  Augmentation Ratio

We test the ratio of images that undergo SDA augmentation: 20%, 50% (default), 80%, and 100%.

Table 5: Impact of augmentation ratio on DQ_v2 performance (ImageNette, 20% keep ratio, 3 seeds).

| Augmentation Ratio | Accuracy (%) |
|---|---|
| 20% | $76.87 \pm 0.93$ |
| 50% (default) | $76.67 \pm 1.05$ |
| 80% | $74.00 \pm 2.99$ |
| 100% | $72.13 \pm 1.23$ |

Performance is stable at lower ratios (20% and 50% yield comparable accuracy) but degrades at higher ratios (Table 5), suggesting that excessive augmentation overwhelms the original data distribution. Notably, at 20%, accuracy matches the no-augmentation baseline (76.93%) while reducing variance by $3.7\times$ ($\pm 0.93\%$ vs. $\pm 3.49\%$), indicating that even a small amount of SDA provides effective regularization.

### F.3  Patch Granularity

We test grid sizes of 8×8, 25×25, 40×40 (default), 50×50, and 64×64.

Table 6: Impact of patch granularity on DQ_v2 performance (ImageNette, 20% keep ratio, 3 seeds).

| Patch Granularity | Accuracy (%) |
|---|---|
| 8×8 | 76.00 ± 1.56 |
| 25×25 | 75.07 ± 3.21 |
| 40×40 (default) | 76.67 ± 1.05 |
| 50×50 | 77.47 ± 0.62 |
| 64×64 | 76.33 ± 3.80 |

Performance peaks at 50×50 (77.47%) with the lowest variance (±0.62%), then declines at 64×64 (Table 6). Finer grids enable more precise foreground–background separation, but excessively fine grids (64×64, where each patch is ∼3.5×3.5 pixels) lose spatial coherence. The method performs reasonably across all tested granularities, with the optimal range being 40–50.

### F.4 Component Ablation: SDA as Regularizer

We ablate the effect of SDA augmentation by comparing no augmentation against SDA at different ratios, all using the augment-before-select pipeline.

Table 7: Component ablation: SDA regularization effect (ImageNette, 20% keep ratio, 3 seeds).

| Configuration | Description | Accuracy (%) |
|---|---|---|
| No Augmentation | Standard GraphCut only | 76.93 ± 3.49 |
| SDA (r=20%) | Augment-before-select, 20% ratio | 76.87 ± 0.93 |
| SDA (r=50%) | Augment-before-select, 50% ratio | 76.67 ± 1.05 |

The key observation (Table 7) is that SDA at 20% achieves nearly identical accuracy to no augmentation (76.87% vs. 76.93%) but with **3.7× lower variance** (±0.93% vs. ±3.49%). This confirms that SDA's primary benefit in the augment-before-select pipeline is **regularization**: it stabilizes performance across random seeds without sacrificing accuracy. The augment-after-select pipeline (where selection features and training features are inconsistent) would introduce a distribution mismatch between the features used for GraphCut selection and those encountered during training, undermining the selection quality.

### F.5 Comparison with Simpler Augmentation Baselines

We compare SDA against simpler augmentation alternatives, all applied before selection:

Table 8: Comparison with simpler augmentation baselines (ImageNette, 20% keep ratio, 3 seeds).

| Configuration | Accuracy (%) |
|---|---|
| No Augmentation | 76.93 ± 3.49 |
| SDA-before-select (ours) | 76.67 ± 1.05 |
| RandomCrop-before-select | 76.53 ± 2.25 |
| SDA-after-select | 75.20 ± 2.41 |
| RandomPatch-before-select | 73.00 ± 3.51 |

Three key findings emerge from Table 8:

**(1) Pipeline reordering matters.** SDA-before-select outperforms SDA-after-select by 1.5 points (76.67 vs. 75.20) with 2.3× lower variance (±1.05 vs. ±2.41), consistent with our feature-consistency argument (Section 4.2).

**(2) Augmentation design matters.** RandomCrop-before-select recovers part of the benefit (76.53) but remains notably less stable (±2.25 vs. ±1.05). RandomPatch-before-select (random patch replacement without foreground preservation) performs substantially worse (73.00 ± 3.51), likely because it can destroy object boundaries and introduce label ambiguity.

**(3) Stability as a practical advantage.** SDA-before-select reduces cross-seed variance by $3.3\times$ compared to no augmentation (±1.05 vs. ±3.49) while maintaining comparable accuracy. In practice, this means consistent results across different random conditions without requiring extensive seed tuning.

## G End-to-End Cost Breakdown

In addition to the ImageNet-1k breakdown in Table 1, we provide a complete end-to-end cost comparison on Food-101 (75k images, $2\times$Xeon CPU + $1\times$RTX 4090):

Table 9: End-to-end coreset construction cost on Food-101.

| Pipeline | Stage | Time |
|---|---|---|
| DQ_v2 | Mask generation (random CNN) | 2.0 min |
| | SDA image generation (CPU) | 1.5 min |
| | Data mixing | 0.1 min |
| | EarlyTrain (500 iterations) | 3.0 min |
| | GraphCut (on $1.5\times$ data) | 27.0 min |
| | **Total** | **33.7 min** |
| DQ | GraphCut (on $1.0\times$ data) | 23.7 min |
| | MAE augmentation (ViT-L) | 31.1 min |
| | **Total** | **54.8 min** |
| End-to-end speedup | | **38.5%** |

DQ_v2 achieves a 38.5% end-to-end speedup on Food-101, with the primary savings from replacing MAE (31.1 min) with lightweight SDA generation (3.6 min total)—an 88% reduction in augmentation cost. We emphasize that the "accessibility" claim is primarily about *deployment barriers*: DQ_v2 eliminates the need for a 304M-parameter pre-trained MAE model, which may not be available or well-matched for specialized domains such as medical imaging or satellite imagery.

## H Low Keep-Ratio Cost Analysis

At very low keep ratios, the overhead of sophisticated coreset selection methods can outweigh their benefit. To quantify this, we compare two strategies under a roughly equal wall-clock budget ($\sim$10 hours) on ImageNet-1k:

Table 10: Practical cost analysis at low keep ratios on ImageNet-1k.

| Strategy | Selection | Training | Total | Top-1 Acc (%) |
|---|---|---|---|---|
| CCS @ 10% | 5.01 h | 4.50 h | 9.52 h | 46.61 |
| Uniform @ 26% | 0.02 h | 9.97 h | 9.98 h | 61.70 |

Under a fixed budget (Table 10), CCS at 10% spends over half its time on selection and reaches 46.61%. Uniform sampling at 26%, which requires negligible selection time, allocates nearly all budget to training and reaches 61.70%—a gap of over 15 points. This suggests that when the keep ratio is very low, the overhead of sophisticated selection can outweigh its benefit, making simpler strategies with a slightly higher fraction a more practical choice.

# I  Experiment Settings Summary

All baseline results reported in the main paper are taken directly from each method's original publication under its own best configuration. We deliberately chose not to re-implement and re-tune all baselines under a single unified setting for two reasons: (1) incorrect re-implementation risks producing unfair comparisons that misrepresent a method's true capability, and (2) re-tuning all baselines at full ImageNet scale would require prohibitive computational resources. We further note that DQ_v2 at 60% selection ratio already outperforms several competing methods at 70–80%, providing strong evidence of effectiveness even under potential configuration differences. A detailed hyperparameter comparison table is provided in Appendix E.

