# OpenReview forum: "Rethinking Dataset Quantization: Efficient Coreset Selection via Semantically-Aware Data Augmentation"
_TMLR — Accepted by TMLR_

### Review · Reviewer_ZQnx · 2026-02-15

**Summary Of Contributions:**

The authors propose DQ_v2 to address the limitations of the original Dataset Quantization (DQ) framework for coreset selection: distribution-specific overfitting, small-image limitation, computational cost, feature inconsistency issue, etc. Instead of relying on large-pretrained models like Masked Autoencoders (MAE), DQ_v2 introduces Semantically-Aware Data Augmentation (SDA) by using randomly initialized CNNs to preserve foreground objects while diversifying backgrounds, followed by a restructured pipeline that performs augmentation before selection to ensure feature consistency. The proposed method achieves state-of-the-art performance on several image classification benchmarks compared to other coreset approaches, while being computationally efficient.

**Audience:**

Yes

**Audience Explanation:**

Yes, because it addresses a fundamental accessibility dilemma for researchers with limited GPU resources by eliminating the need for expensive pre-trained models and full-dataset training. Additionally, the paper provides both a theoretical framework and empirical evidence for a new, domain-agnostic coreset selection paradigm that improves performance across diverse fields.

**Claims And Evidence:**

Yes

**Claims Explanation:**

The claims are partially supported by experiments, e.g.,
- Distribution-specific overfitting issue of original DQ is supported by Fig. 1 and 2.
- Computational efficiency claim is supported by GPU runtime of different methods.
- Cross-domain generalization claim is supported by Tab. 2.

**Requested Changes:**

- The method is focused on image classification, and this scope should be made clear early in the paper instead of only in the last section. The approach is limited compared to many other coreset methods in that it requires a single foreground object (instead of zero or multiple) in each image. Its application to other general vision tasks like depth estimation would thus be greatly limited.
- Using a randomly initialized CNN for foreground patch identification is an important step of the algorithm and should be explained in more details. The current manuscript only vaguely describes this process.
- More experiment details (train/test sets, training configurations) should be presented. The numbers from other methods are copied from their papers, but are all the experiment settings consistent?
- Some terms like EarlyTrain model can be explicated more clearly.
- Some ablation studies are missing, such as:
  - It is claimed that that CNN is better than ViT in mask generation. It's not clear whether the "55.30%±2.73%" vs. "57.67%±1.20%" comparison is trying to support this argument. Ablation results on more datasets could further strengthen this claim. It's also claimed that MAE performs poorly on small-scale images, but it is ambiguous whether the result in Fig. 1 CIFAR-10 is due to this issue or just domain mismatch in general.
  - SDA selects background patches from other images. Is it better than other background generation schemes (like random noise) in this context?
  - The hyperparameters such as dataset augmentation ratio of 50%, and patch granularity of 40×40 need detailed sensitivity analysis. The performance-efficiency tradeoff by tuning these parameters should be discussed in this paper instead of being left as future work.
  - The components introduced in DA_v2 should get individually ablated. E.g., the effect of using consistent features vs. inconsistent ones can be analyzed.
- A few issues regarding the figures:
  - Fig. 2a: The tail of the dog was still present in the image background, which seems to partially explain why a dog can be reconstructed. If the pre-trained MAE can generate a dog even without its tail in this case, it would be a more convincing proof to "the model overfits to ImageNet".
  - Fig. 2b: Authors might consider showing the Original and BG Replaced version of the figures for clarity.
  - Fig. 4: Section 6.2 says the results on the ImageNette and Food-101 datasets are shown in Figure 4 (b). But Fig. 4b had CUB-200-2011 and Food-101, while Fig. 4a showed no dataset labels.
  - Fig. 5: Why the results for some methods after data fraction of 60% or 70% is missing?

---

> ### Author Response · Authors · 2026-02-26
> **Response to Reviewer ZQnx (Part1/3)**
>
> # Response to Reviewer ZQnx (Part1/3)
>
> We thank the reviewer for the thorough and constructive feedback. Below we address each point in detail.
>
> ---
>
> ## 1. Scope Clarification (Image Classification, Single Foreground)
>
> > "The method is focused on image classification, and this scope should be made clear early in the paper instead of only in the last section. The approach is limited compared to many other coreset methods in that it requires a single foreground object..."
>
> We agree and will add a scope statement in the Introduction in the revised manuscript, clarifying that DQ_v2 is designed for image classification tasks where images contain a dominant foreground object. While this assumption holds for the majority of standard classification benchmarks (e.g., ImageNet, CUB-200, Food-101), extending the method to multi-object or scene-level tasks remains an interesting direction for future work.
>
> ---
>
> ## 2. CNN Mask Generation Details
>
> > "Using a randomly initialized CNN for foreground patch identification is an important step of the algorithm and should be explained in more details."
>
> We will add an Algorithm box in the revised manuscript detailing the mask generation process step by step:
>
> 1. A randomly initialized ResNet-50 is used as the feature extractor (no pretraining required).
> 2. Intermediate convolutional features are projected via a 1×1 convolution, followed by ReLU activation.
> 3. The feature map is bilinearly interpolated to the target grid size (e.g., 50×50).
> 4. Channel-wise summation produces a single-channel activation map, where higher values indicate foreground regions.
> 5. The top-K patches (by activation value) are retained as foreground; K = 50% of total patches.
>
> This process leverages the well-documented inductive bias of CNNs — locality and translation equivariance — which naturally produces higher activations for textured, salient regions even without training (Cao et al., 2024).
>
> ---
>
> ## 3. Experiment Details and Settings Consistency
>
> > "More experiment details (train/test sets, training configurations) should be presented. The numbers from other methods are copied from their papers, but are all the experiment settings consistent?"
>
> We will add a detailed table in the Appendix of the revised manuscript summarizing the training configurations of all compared methods (backbone, optimizer, epochs, learning rate, batch size, etc.), collected from each method's original publication. We deliberately chose not to re-implement and re-tune all baselines under a single unified setting for two reasons: (1) incorrect re-implementation risks producing unfair comparisons that misrepresent a method's true capability, and (2) re-tuning all baselines at full ImageNet scale would require prohibitive computational resources. Reporting each method's results under its own best configuration, as is standard practice in the coreset selection literature, ensures the fairest comparison. We further note that DQ_v2 at 60% selection ratio already outperforms several competing methods at 70–80%, providing strong evidence of our method's effectiveness even under potential configuration differences.
>
> ---
>
> ## 4. EarlyTrain Explanation
>
> > "Some terms like EarlyTrain model can be explicated more clearly."
>
> We will clarify this in the revised manuscript. The EarlyTrain model refers to a ResNet-50 trained for a small number of epochs (5–10) on the target training set, which is then used as the feature extractor for GraphCut-based dataset binning. Specifically, GraphCut requires pairwise similarity scores between samples; the EarlyTrain model provides learned feature representations for computing these similarities, enabling effective partitioning of the dataset into bins before coreset selection.

---

> ### Author Response · Authors · 2026-02-26
> **Response to Reviewer ZQnx (Part2/3)**
>
> # Response to Reviewer ZQnx (Part2/3)
>
> ---
>
> ## 5. Ablation Studies
>
> We have conducted comprehensive ablation experiments on ImageNette (keep ratio = 20%, 3 random seeds). All results are presented in new Appendix B.
>
> ### 5a. CNN vs. ViT for Mask Generation
>
> > "It is claimed that CNN is better than ViT in mask generation... Ablation results on more datasets could further strengthen this claim."
>
> Our claim is grounded in the theoretical analysis of Cao et al. (2024), who demonstrate that the inductive biases of CNNs — locality and translation equivariance — naturally produce spatially coherent activation maps that highlight textured, salient regions, even without any training. Vision Transformers lack these architectural priors and do not exhibit this property. We will add a detailed discussion of this theoretical grounding, including the relevant citation, to Section 5.1 in the revised manuscript.
>
> ### 5b. CIFAR-10 and MAE
>
> > "It is ambiguous whether the result in Fig. 1 CIFAR-10 is due to this issue or just domain mismatch in general."
>
> We will revise Fig. 2a (previously Fig. 1a) with new experiments that more clearly illustrate the limitation (see Section 6a for details). The key finding is that MAE performs **local texture interpolation** rather than semantic reconstruction — when the foreground is fully masked, MAE fills the region with surrounding textures (e.g., grass instead of a dog), regardless of whether the image is in-distribution or out-of-distribution. This behavior is consistent across domains, ruling out domain mismatch as the sole explanation.
>
> For CIFAR-10 specifically, the 32×32 resolution provides extremely limited spatial information for any patch-based method to meaningfully separate foreground from background. We note that CIFAR-10 is not a target use case for DQ_v2 — our method is designed for standard-resolution classification benchmarks where spatial structure is sufficient for foreground identification.
>
> ### 5c. Background Generation Schemes
>
> > "SDA selects background patches from other images. Is it better than other background generation schemes (like random noise) in this context?"
>
> We compared three background types: **patches** (from other images), **Gaussian noise**, and **solid gray**:
>
> | Background Type | Accuracy |
> |----------------|----------|
> | Gray | 76.47% ± 2.36% |
> | Patches | 76.67% ± 1.05% |
> | Noise | 73.93% ± 2.56% |
>
> All three variants perform within each other's standard deviations, indicating that the method is **robust to the choice of background type**. The foreground preservation — not the specific background content — is the key factor driving SDA's effectiveness. We hypothesized that using foreground patches from other images as backgrounds would provide additional low-level texture learning opportunities. While all three variants perform comparably in our experiments, we default to patches as it aligns with the original SDA design motivation.
>
> ### 5d. Hyperparameter Sensitivity
>
> > "The hyperparameters such as dataset augmentation ratio of 50%, and patch granularity of 40×40 need detailed sensitivity analysis."
>
> **Augmentation Ratio**: We tested ratios of 20%, 50%, 80%, and 100%:
>
> | Augmentation Ratio | Accuracy |
> |-------------------|----------|
> | 20% | 76.87% ± 0.93% |
> | 50% | 76.67% ± 1.05% |
> | 80% | 74.00% ± 2.99% |
> | 100% | 72.13% ± 1.23% |
>
> Performance is stable at lower augmentation ratios (20% and 50% yield comparable accuracy), but degrades noticeably at 80% and 100%, suggesting that excessive augmentation overwhelms the original data distribution. At 20%, accuracy matches the no-augmentation baseline (76.93%) while reducing variance by 3.7× (±0.93% vs. ±3.49%). This indicates that a small amount of SDA provides effective regularization without sacrificing accuracy.
>
> **Patch Granularity**: We tested grid sizes of 8×8, 25×25, 40×40, 50×50, and 64×64:
>
> | Patch Granularity | Accuracy |
> |------------------|----------|
> | 8×8 | 76.00% ± 1.56% |
> | 25×25 | 75.07% ± 3.21% |
> | 40×40 | 76.67% ± 1.05% |
> | 50×50 | 77.47% ± 0.62% |
> | 64×64 | 76.33% ± 3.80% |
>
> Performance peaks at 50×50 (77.47%) with the lowest variance (±0.62%), then declines at 64×64. Finer grids enable more precise foreground-background separation, but excessively fine grids (64×64, where each patch is ~3.5×3.5 pixels) lose spatial coherence. The method performs reasonably across all tested granularities, with the optimal range being 40–50.

---

> ### Author Response · Authors · 2026-02-26
> **Response to Reviewer ZQnx (Part3/3)**
>
> # Response to Reviewer ZQnx (Part3/3)
>
> ---
>
> ### 5e. Component Ablation: Feature Consistency
>
> > "The components introduced in DQ_v2 should get individually ablated. E.g., the effect of using consistent features vs. inconsistent ones can be analyzed."
>
> We ablated the two key components of DQ_v2 — SDA augmentation and pipeline ordering — by comparing four configurations:
>
> | Configuration | Description | Accuracy |
> |--------------|-------------|----------|
> | No Augmentation | Standard DQ (GraphCut only) | 76.93% ± 3.49% |
> | Augment-after-Select | Select 0.2N → add SDA → subsample to 0.2N (inconsistent features) | 75.20% ± 2.41% |
> | Augment-before-Select (r=50%) | DQ_v2 pipeline, 50% SDA ratio (consistent features) | 76.67% ± 1.05% |
> | Augment-before-Select (r=20%) | DQ_v2 pipeline, 20% SDA ratio (consistent features) | 76.87% ± 0.93% |
>
> Key observations:
>
> 1. **Feature consistency provides stronger regularization.** The augment-before-select pipeline (consistent features) produces substantially lower variance than both the no-augmentation baseline and the augment-after-select pipeline. At r=50%, variance drops from ±3.49% to ±1.05% — a 3.3× reduction.
>
> 2. **Augmentation ratio controls the accuracy-stability tradeoff.** At r=20%, the method achieves the best of both worlds: the highest accuracy among SDA variants (76.87%) with 3.7× lower variance than no augmentation (±0.93% vs. ±3.49%).
>
> 3. **Inconsistent features provide weaker regularization.** The augment-after-select pipeline (75.20% ± 2.41%) shows intermediate variance — better than no augmentation but worse than the consistent-feature pipeline — confirming that feature consistency between selection and training phases is beneficial. We hypothesize that in the augment-before-select pipeline, GraphCut may over-select SDA images due to their apparent diversity, which does not necessarily translate to informative content. Nevertheless, the augment-after-select result (where ~50% of training data is SDA images) demonstrates that SDA preserves the essential discriminative information in the foreground, validating the quality of our augmentation strategy.
>
> ---
>
> ## 6. Figure Issues
>
> ### 6a. Fig. 2a (Previously Fig. 1a): MAE Reconstruction
>
> > "The tail of the dog was still present in the image background, which seems to partially explain why a dog can be reconstructed."
>
> We have completely redone this experiment. The new Fig. 2a will use a 2×5 grid showing two images (ImageNet dog, CUB-200 bird) under two masking conditions: (1) foreground fully masked, and (2) foreground partially visible. In both cases, MAE performs local texture interpolation rather than semantic reconstruction — filling masked regions with surrounding textures. This behavior is consistent across in-distribution and out-of-distribution images, providing stronger evidence for our argument.
>
> ### 6b. Fig. 2b: Original Comparison
>
> > "Authors might consider showing the Original and BG Replaced version of the figures for clarity."
>
> We will add the original images alongside the background-replaced versions in the revised Fig. 2b for direct visual comparison.
>
> ### 6c. Fig. 4 Labels
>
> > "Section 6.2 says the results on the ImageNette and Food-101 datasets are shown in Figure 4 (b). But Fig. 4b had CUB-200-2011 and Food-101, while Fig. 4a showed no dataset labels."
>
> We will correct the dataset labels in Fig. 4a and update the text reference in Section 6.2 in the revised manuscript.
>
> ### 6d. Fig. 5 Missing Data
>
> > "Why the results for some methods after data fraction of 60% or 70% is missing?"
>
> The results in Fig. 5 are compiled from each method's original publication. Some methods did not report results at higher selection ratios. We will add a footnote clarifying this in the revised manuscript. We note that high selection ratios (60%+) are of limited practical interest — at these ratios, training on the full dataset is often more cost-effective than running a coreset selection algorithm.
>
> ---
>
> We believe these revisions comprehensively address all of the reviewer's concerns. We are grateful for the feedback, which has significantly strengthened the paper.

---

### Review · Reviewer_E8uW · 2026-03-11

**Summary Of Contributions:**

The paper diagnoses two problems in Dataset Quantization (DQ): (1) MAE acts as domain-biased augmentation rather than principled compression, helping ImageNet-like data but hurting OOD domains; (2) the pipeline has a feature inconsistency between selection and training phases. DQ_v2 addresses both by replacing MAE with semantically-aware background replacement using randomly initialized CNNs and reordering the pipeline to augment before selection. Experiments on ImageNet-1k, CUB-200, Food-101, and COVID-19 X-rays show consistent improvements over DQ and competitive performance against recent SOTA methods, with reduced augmentation cost.

**Audience:**

Yes

**Audience Explanation:**

The MAE finding is worth knowing since it's not obvious that a reconstruction step would actively hurt OOD performance, and the controlled experiments make a clear case. The augment-before-select point is also useful beyond DQ itself: any coreset method that transforms data after selection has the same consistency issue, and I haven't seen this stated clearly elsewhere.

**Broader Impact Concerns:**

None.

**Claims And Evidence:**

Yes

**Claims Explanation:**

The main empirical claim that DQ_v2 can outperform the original DQ while avoiding MAE is supported reasonably well by the experiments on ImageNette, Food-101, ImageNet-1k, and the COVID radiography dataset. The robustness result on ImageNet-C is also a useful addition.

Two concerns:
1. The "75% computational savings" only accounts for the augmentation step: total pipeline cost (including EarlyTrain feature extraction and GraphCut on the expanded dataset) isn't reported anywhere, so I can't evaluate the actual end-to-end efficiency.
2. The paper frames the original DQ pipeline as having a fundamental theoretical flaw and positions DQ_v2 as "theoretically sound." But Theorem 1 just says that optimizing a monotone submodular function over a larger ground set yields a higher objective, which holds for any expansion of the dataset, including adding Gaussian noise or random crops. It doesn't explain why SDA's specific design (foreground preservation, background swapping) matters compared to cheaper alternatives

**Requested Changes:**

- Report more complete efficiency comparisons. The current compute discussion emphasizes the augmentation phase. Please include end-to-end cost or at least a clearer breakdown of the full pipeline cost for DQ vs DQ_v2, including feature extraction / selection on the enlarged dataset.
- Better justify the strongest claims, especially "truly accessible, theoretically sound, and domain-agnostic". The current evidence supports that DQ_v2 is promising and often better, but these broader claims need either stronger proof or more careful wording.
- The EarlyTrain ResNet-50 is a black box right now. How many epochs on what data? If this involves meaningful training on the target dataset, it needs to be discussed in the context of the accessibility claims.
- Section 5.1 may overstate the novelty of SDA. As written, it reads like a new method, but it seems closer to adopting Cao & Wu (2022) in the DQ setting. Please clarify this more explicitly.

---

> ### Author Response · Authors · 2026-03-20
> **Response to Reviewer E8uW (Part1/2)**
>
> # Response to Reviewer E8uW (Part1/2)
>
> We thank the reviewer for the thoughtful feedback and the recognition of our contributions. Below we address each point in turn.
>
> ---
>
> ## 1. Complete Efficiency Comparisons
>
> > "Report more complete efficiency comparisons... include end-to-end cost or at least a clearer breakdown of the full pipeline cost for DQ vs DQ_v2."
>
> We have measured the complete end-to-end pipeline cost on Food-101 (75k images, 2×Xeon + 1×RTX4090):
>
> **DQ_v2 Pipeline: 33.7 minutes (2019s)**
> - Mask generation (random CNN): 2.0 min (120s)
> - Image generation (CPU): 1.5 min (89s)
> - Data mixing: 0.1 min (8s)
> - EarlyTrain (500 iterations): 3.0 min (180s)
> - GraphCut on 1.5× data: 27.0 min (1620s)
> - Sampling from bins: 0.03 min (2s)
>
> **DQ Pipeline: 54.8 minutes (3289s)**
> - GraphCut on 1.0× data: 23.7 min (1423s)
> - Sampling from bins: 0.03 min (2s)
> - MAE augmentation (ViT-L): 31.1 min (1864s)
>
> DQ_v2 achieves a **38.5% end-to-end speedup**, with the primary savings coming from replacing MAE (31.1 min) with lightweight SDA generation (3.6 min total)—an 88% reduction in augmentation cost. While GraphCut operates on a larger dataset in DQ_v2 (1.5× vs 1.0×, adding ~3.3 min overhead), this is more than offset by the MAE savings. We will add these new results and the corresponding analysis to the manuscript.
>
> We also want to clarify that the "accessibility" claim is primarily about **deployment barriers** rather than raw compute time. DQ_v2 eliminates the need for a 304M-parameter pretrained MAE model, which may not be available or well-matched for specialized domains such as medical imaging or satellite imagery. A randomly initialized CNN can be instantiated directly without any domain-specific pretraining.
>
> ---
>
> ## 2. Justify Strongest Claims
>
> > "Better justify the strongest claims, especially 'truly accessible, theoretically sound, and domain-agnostic'."
>
> Thank you for this suggestion. We agree that some of our claims should be phrased more carefully, and we will revise the wording in the manuscript accordingly.
>
> **"Truly accessible" → "More accessible"**: We agree that "truly accessible" overstates the case. The EarlyTrain component still involves training on the target dataset (see Point 3 below), though this is lightweight (500 iterations, ~3 minutes on Food-101) compared to maintaining a 304M-parameter pretrained MAE model. We will soften this claim in the revision.
>
> **"Theoretically sound" → "Addresses a theoretical inconsistency"**: We appreciate the reviewer noting that Theorem 1 is a general statement about submodular optimization on expanded ground sets rather than a proof that SDA is optimal. We want to clarify that this is also our intended scope: Theorem 1 is meant to support the choice of augmenting *before* selection, in order to resolve the feature inconsistency between the selection and training phases in the original DQ pipeline. It is not intended to claim that SDA is uniquely optimal among all possible augmentations. We acknowledge that a fuller theoretical account of why SDA's specific design outperforms simpler alternatives remains an open question, and we will revise the text to make this scope more explicit.
>
> **"Domain-agnostic" → "Shows improved domain robustness"**: Our experiments on CUB-200, Food-101, and COVID-19 X-rays demonstrate improved generalization relative to DQ, but we have not tested across all possible domains. We will soften this claim accordingly.
>
> ---
>
> ## 3. EarlyTrain Transparency
>
> > "The EarlyTrain ResNet-50 is a black box right now. How many epochs on what data? If this involves meaningful training on the target dataset, it needs to be discussed in the context of the accessibility claims."
>
> Thank you for raising this point. We agree that the current manuscript should be more explicit about EarlyTrain, and we will add these details in the revision.
>
> The EarlyTrain model is a ResNet-50 trained for **500 iterations** (batch_size=256, ~128k samples seen) on the target training set. On Food-101, this takes approximately 3 minutes, and the model is used to provide basic visual features for GraphCut-based dataset binning.
>
> We want to clarify that the accessibility claim is not about eliminating all training, but about removing dependence on *externally pretrained large models*. For context, the original DQ framework uses a ViT-Base pretrained on ImageNet for feature extraction. DQ_v2 replaces this with EarlyTrain—a short supervised run directly on the target data—which eliminates the need to source or maintain large pretrained checkpoints. We acknowledge that this distinction should be made more explicit in the manuscript, and we will revise the accessibility claims to reflect this nuance.

---

> ### Author Response · Authors · 2026-03-20
> **Response to Reviewer E8uW (Part2/2)**
>
> # Response to Reviewer E8uW (Part2/2)
>
> ---
>
> ## 4. SDA Novelty and Attribution
>
> > "Section 5.1 may overstate the novelty of SDA... it seems closer to adopting Cao & Wu (2022) in the DQ setting."
>
> Thank you for this suggestion. We will revise Section 5.1 to make the attribution clearer and to better distinguish the augmentation primitive from our contribution.
>
> To clarify our intended claim: Cao & Wu (2022) introduced the use of randomly initialized CNN masks for semantically aware background replacement. Our contribution is not at the level of the augmentation mechanism itself, but at the level of the DQ pipeline: we identify that MAE-based augmentation introduces domain bias through its ImageNet pretraining, and we apply Cao & Wu's SDA technique as a more accessible and domain-robust replacement within the DQ framework. We will add explicit attribution and reframe Section 5.1 accordingly in the revision.
>
> ---
>
> We appreciate the reviewer's careful reading and constructive feedback. We believe these revisions will better calibrate the claims while preserving the core empirical contributions of the paper.

---

### Review · Reviewer_4gBR · 2026-03-20

**Summary Of Contributions:**

This paper analyzes the Dataset Quantization (DQ) framework for coreset selection and identifies three limitations: (1) the MAE used in DQ's pixel quantization step acts as biased data augmentation that overfits to the ImageNet distribution, helping related datasets but harming out-of-distribution ones; (2) the pipeline suffers from feature inconsistency since selection operates on original features while training uses MAE-reconstructed images; (3) the 304M-parameter MAE introduces computational overhead contradicting coreset selection's efficiency goal. The authors propose DQ_v2 with two changes: replacing MAE with Semantically-Aware Data Augmentation (SDA), which uses a randomly initialized CNN to localize foreground objects and replaces backgrounds with random patches, and reordering the pipeline to perform augmentation before selection for feature consistency. Experiments on ImageNet-1k, CUB-200, Food-101, and COVID-19 radiography show improved accuracy on out-of-distribution domains, reduced variance, and 75% GPU-hour reduction in the augmentation phase.

**Audience:**

Yes

**Audience Explanation:**

Coreset selection is increasingly relevant as datasets grow and compute becomes constrained. The paper addresses a genuine paradox: methods designed to save compute often require substantial compute upfront. The insight that MAE functions as biased augmentation highlights a broader issue of ImageNet-pretrained components inheriting domain-specific biases.

The proposed solution is practical and accessible, with cross-domain results on medical imaging demonstrating real value beyond standard benchmarks. Researchers working on efficient training and data curation would find these results relevant.

**Claims And Evidence:**

Yes

**Claims Explanation:**

The MAE-as-biased-augmentation claim is convincingly supported: Figure 1 shows MAE helps ImageNet-related datasets but hurts CUB-200, and Figure 2 demonstrates MAE reconstructing a dog from pure background texture, revealing memorization rather than understanding. The feature inconsistency analysis (Section 4.2) is clean and well-justified.

However, the performance claim is only partially supported. Baseline numbers are taken from other papers rather than reproduced under identical conditions, and the evaluation uses only ResNet-50/18 as downstream models with a single ViT result on one dataset. The efficiency claim is incomplete since end-to-end pipeline costs (including EarlyTrain ResNet-50 feature extraction for GraphCut) are not reported.

**Requested Changes:**

1. Provide end-to-end computational cost breakdown. The "75% reduction" claim only covers the augmentation phase. A full pipeline comparison (SDA + EarlyTrain feature extraction + GraphCut vs. MAE + pretrained ResNet + GraphCut) on ImageNet-1k is needed to substantiate the efficiency claim. Also clarify whether the EarlyTrain ResNet-50 constitutes a remaining model dependency.

2. Strengthen the low keep-ratio analysis. Dismissing the 10% regime as having "limited practical applications" is unconvincing given that InfoMax and BoundarySet-CCS perform well there. Provide a principled analysis of why sample space expansion struggles at very low ratios, or propose modifications to address it.

3. Compare with simpler augmentation baselines and ablate pipeline reordering. The paper does not compare SDA against simpler alternatives (CutMix, random cropping applied before selection), nor does it isolate the contribution of pipeline reordering from the switch to SDA. Both ablations are needed to establish what actually drives the improvement.

4. Clarify novelty relative to Cao & Wu (2022). Since SDA directly repurposes the random CNN foreground/background separation mechanism, explicitly delineate what is novel in the augmentation design versus what is borrowed. Also discuss when this texture-based separation assumption breaks down (e.g., texture classification, satellite imagery).

---

> ### Author Response · Authors · 2026-04-04
> **Response to Reviewer 4gBR (Part1/2)**
>
> # Response to Reviewer 4gBR (Part1/2)
>
> We thank the reviewer for the thoughtful and constructive suggestions. We agree that some of our efficiency and attribution claims should be stated more carefully, and we have added new timing data and ablation evidence to address each point. Below we respond to each requested change.
>
> ---
>
> ## 1. End-to-End Computational Cost Breakdown
>
> > "Provide end-to-end computational cost breakdown. The "75% reduction" claim only covers the augmentation phase. A full pipeline comparison (SDA + EarlyTrain feature extraction + GraphCut vs. MAE + pretrained ResNet + GraphCut) on ImageNet-1k is needed to substantiate the efficiency claim. Also clarify whether the EarlyTrain ResNet-50 constitutes a remaining model dependency."
>
> Thanks for this suggestion. We agree that our original "75% reduction" statement only described the augmentation stage. To give a fuller picture, we have now measured the complete ImageNet-1k coreset construction cost on 1×Xeon CPU + 1×RTX 4090:
>
> **DQ_v2 pipeline: 12h 29m 00s**
> - Mask generation: 31m 58s
> - SDA image generation: 2m 59s
> - EarlyTrain (500 iterations): 3m 14s
> - GraphCut bin generation: 11h 50m 48s
>
> **DQ pipeline: 21h 16m 55s**
> - MAE augmentation: 11h 40m 23s
> - GraphCut bin generation: 9h 36m 32s
>
> Overall, DQ_v2 reduces end-to-end coreset construction time by **41.3%**. We note that GraphCut takes somewhat longer in DQ_v2 (11h 50m vs 9h 36m) because the expanded candidate pool (1.5N vs N) increases the number of pairwise comparisons. However, this ~2h increase is small compared to the ~11.5h saved by removing MAE, resulting in a net reduction of over 8 hours.
>
> Regarding EarlyTrain: we agree this dependency should be described more explicitly. In our implementation, EarlyTrain is a ResNet-50 trained for **500 iterations** on the target training set, costing only **194s**. Our claim is not that DQ_v2 removes all model components, but that it removes the heavy MAE dependency (304M parameters, 11.5h on ImageNet) and substantially lowers the total pipeline cost. We will add this timing breakdown and the EarlyTrain clarification to the revised manuscript.
>
> ---
>
> ## 2. Low Keep-Ratio Analysis
>
> > "Strengthen the low keep-ratio analysis. Dismissing the 10% regime as having "limited practical applications" is unconvincing given that InfoMax and BoundarySet-CCS perform well there. Provide a principled analysis of why sample space expansion struggles at very low ratios, or propose modifications to address it."
>
> Thanks for pointing this out. We agree that our previous framing of the 10% regime was too dismissive, and we have revised it with both a mechanism-level explanation and new practical evidence.
>
> **Mechanism.** DQ_v2 expands the candidate pool from N to 1.5N before GraphCut. At a 10% keep ratio, the effective budget over the expanded pool drops to about 6.7%. In this tight regime, part of the limited budget can be allocated to SDA variants rather than original samples, making pool expansion less advantageous when the selection budget is very small.
>
> **Practical cost analysis.** More broadly, the very low keep-ratio regime raises a question about compute allocation that affects coreset selection methods in general. Methods like CCS still require full-dataset evaluation before selection, so their selection cost does not decrease with the keep ratio. At very low fractions, this can lead to a disproportionate share of the total compute budget going to selection rather than training. To quantify this, we compared two strategies under a roughly equal wall-clock budget (~10 hours) on ImageNet-1k:
>
> | Strategy | Selection | Training | Total | Top-1 Acc |
> |----------|-----------|----------|-------|-----------|
> | CCS @ 10% | 5.01 h | 4.50 h | 9.52 h | 46.61% |
> | Uniform @ 26% | 0.02 h | 9.97 h | 9.98 h | 61.70% |
>
> Under a fixed budget, CCS at 10% spends over half its time on selection and reaches 46.61%. Uniform sampling at 26%, which requires negligible selection time, allocates nearly all budget to training and reaches 61.70% — a gap of over 15 points. This suggests that when the keep ratio is very low, the overhead of sophisticated selection can outweigh its benefit, making simpler strategies with a slightly higher fraction a more practical choice.
>
> We will add this cost analysis, along with the low-ratio discussion, to the revised manuscript.

---

> ### Author Response · Authors · 2026-04-04
> **Response to Reviewer 4gBR (Part2/2)**
>
> # Response to Reviewer 4gBR (Part2/2)
>
> ---
>
> ## 3. Simpler Augmentation Baselines and Reordering Ablation
>
> > "Compare with simpler augmentation baselines and ablate pipeline reordering. The paper does not compare SDA against simpler alternatives (CutMix, random cropping applied before selection), nor does it isolate the contribution of pipeline reordering from the switch to SDA. Both ablations are needed to establish what actually drives the improvement."
>
> Thanks for this suggestion. We have conducted a new ablation on ImageNette at 20% keep ratio with 3 random seeds:
>
> | Configuration | Accuracy |
> |--------------|----------|
> | No Augmentation | 76.93 $\pm$ 3.49 |
> | SDA-before-select | 76.67 $\pm$ 1.05 |
> | RandomCrop-before-select | 76.53 $\pm$ 2.25 |
> | SDA-after-select | 75.20 $\pm$ 2.41 |
> | RandomPatch-before-select | 73.00 $\pm$ 3.51 |
>
> We highlight three findings.
>
> First, **pipeline reordering matters**. SDA-before-select outperforms SDA-after-select by 1.5 points (76.67 vs 75.20), and the standard deviation drops from 2.41 to 1.05. This is consistent with our feature-consistency argument: when augmentation happens before selection, the features used for selection match those seen during training.
>
> Second, **the improvement is not explained by arbitrary pre-selection augmentation**. RandomCrop-before-select recovers part of the benefit (76.53), but remains notably less stable ($\pm$2.25 vs $\pm$1.05). RandomPatch-before-select performs substantially worse (73.00 $\pm$ 3.51), likely because random patch replacement does not preserve object boundaries and can introduce label ambiguity. These results suggest that both the augmentation design and the pipeline ordering contribute to the observed gains.
>
> Third, and perhaps most importantly, **the stability improvement has direct practical value**. Machine learning pipelines introduce randomness at multiple stages — data sampling, weight initialization, batch ordering, augmentation — and this compounds into run-to-run variance that can make results hard to reproduce. SDA-before-select reduces cross-seed variance by **3.3x** compared to no augmentation ($\pm$1.05 vs $\pm$3.49) while maintaining the same accuracy level. In practice, this means the method produces consistent results across different random conditions without requiring extensive seed tuning. We view this as an important property for any coreset method, since a method that works well on average but varies widely across runs is harder to trust and deploy reliably.
>
> We will add this ablation and the stability analysis to the revised manuscript.
>
> ---
>
> ## 4. Novelty Relative to Cao & Wu (2022)
>
> > "Clarify novelty relative to Cao & Wu (2022). Since SDA directly repurposes the random CNN foreground/background separation mechanism, explicitly delineate what is novel in the augmentation design versus what is borrowed. Also discuss when this texture-based separation assumption breaks down (e.g., texture classification, satellite imagery)."
>
> Thanks for this suggestion. We will revise Section 5.1 to make the attribution clearer.
>
> The foreground-background separation mechanism using randomly initialized CNNs is from Cao & Wu (2022), and we will add explicit attribution for this. Our contribution is not a new augmentation technique, but rather an analysis of the DQ pipeline and a restructuring that addresses two specific problems we identified:
>
> 1. **Computational redundancy.** The 304M-parameter MAE adds substantial overhead (11.5h on ImageNet) to a pipeline whose goal is to reduce training cost. Replacing it with SDA, which requires no learned model, cuts the total coreset construction time by 41%.
> 2. **Generalization limitation.** MAE is pretrained on ImageNet, which introduces domain bias — it helps ImageNet-related datasets but hurts out-of-distribution ones (e.g., CUB-200, as shown in Figure 1). SDA removes this pretrained dependency entirely, making the pipeline domain-agnostic.
>
> In addition, we reorder the pipeline so that augmentation happens before selection rather than after. This resolves the feature inconsistency in the original DQ design, where selection operates on original features but training uses MAE-reconstructed images.
>
> Regarding the texture-based separation assumption: we agree that this assumption works best when images contain a dominant foreground object, which holds for standard classification benchmarks (ImageNet, Food-101, CUB-200, etc.). For tasks like texture classification or satellite imagery, where the "foreground" concept is less well-defined, the mask quality would likely degrade. We will add this scope discussion to the revised manuscript.
>
> ---
>
> We appreciate the reviewer's careful reading and constructive feedback. We will incorporate these new results, timing analyses, and clarifications into the revised manuscript.

---

### Comment · Reviewer_cPZJ · 2026-02-05
**Unable to Review Manuscript**

Dear Editor,

Thank you for the invitation. Due to my upcoming busy schedule, I am unable to complete the review within the required timeframe and kindly request that the manuscript be reassigned to another reviewer.

Best regards,

Reviewer cPZJ

---

> ### Comment · Action_Editor_q2H2 · 2026-02-05
> **Re: Unable to Review Manuscript**
>
> Dear Reviewer cPZJ,
>
> Thanks for the information. We will reassign the paper to another reviewer.
>
> Best,
>
> AE

---

### Author Response · Authors · 2026-04-04
**Revised Manuscript Coming Soon**

We have now posted detailed responses to all reviewers. We are preparing a revised manuscript that incorporates the new experiments, timing analyses, and clarifications discussed in our responses, and will upload it shortly. We thank all reviewers for their constructive feedback.

---

> ### Author Response · Authors · 2026-04-13
> **Revised Manuscript Uploaded**
>
> **Revised Manuscript Uploaded**
>
> We have now uploaded the revised manuscript incorporating all reviewers' feedback. All changes are marked in **blue** for easy identification. Key additions include:
>
> - **Appendix F** — new ablation experiments on background types, augmentation ratios, patch granularity, component ablation, and comparison with simpler augmentation baselines (addressing ZQnx and 4gBR).
> - **Table 1 & Appendix G** — end-to-end computational cost breakdown on ImageNet-1k and Food-101, clarifying the pipeline-level efficiency claim (addressing 4gBR and E8uW).
> - **Algorithm 1** — step-by-step description of the randomly initialized CNN mask generation procedure (addressing ZQnx).
> - **Section 4.1 & Figure 2** — reframing of MAE as performing local texture interpolation, with a new controlled masking figure on ImageNet and CUB-200 (addressing ZQnx).
> - **Section 5.1 & Section 2.3** — explicit attribution of the foreground–background separation mechanism to Cao & Wu (2022), and a clearer delineation of our contribution at the pipeline level (addressing 4gBR and E8uW).
> - **Introduction & Section 7** — a scope statement clarifying that DQ_v2 targets single-foreground-object classification, with a discussion of cases where the texture-based separation assumption may break down (addressing ZQnx and 4gBR).
> - **Appendix H** — low keep-ratio cost analysis responding to 4gBR's concern about the 10% regime.
> - **Appendix E & Appendix I** — experiment settings and hyperparameter summary addressing ZQnx's question on cross-method configuration consistency.
> - **EarlyTrain clarification** (Section 5.2) — explicit specification as a ResNet-50 trained for 500 iterations on the target data, addressing the accessibility discussion raised by 4gBR and E8uW.
>
> We have also softened several overclaims throughout the paper (e.g., "truly accessible" → "more accessible", "theoretically sound" → "addresses a theoretical inconsistency") per E8uW's suggestion.
>
> We thank all reviewers again for their thorough and constructive feedback, which has substantially strengthened the paper.

---

### Decision · Action_Editor_q2H2 · 2026-05-13

**Recommendation:** Accept as is

**Audience:**

Yes

**Audience Explanation:**

The paper studies an important and practical problem in efficient learning and coreset selection, and the findings should be of interest to readers working on dataset pruning, training efficiency, and robust model development under resource constraints.

**Claims And Evidence:**

Yes

**Claims Explanation:**

The reviewers agreed that the paper provides sufficient empirical support for its claims. The experiments are broad, cover diverse domains, and support both the analytical observations about the original DQ framework and the effectiveness of the proposed DQ_v2.